Submission

HEPHY-ML-25-03

# Higgs Signal Strength Estimation with Machine Learning under Systematic Uncertainties

**Minxuan He[1], Claudius Krause[2], and Daohan Wang[2]**

**1** Academy of Mathematics and Systems Science, Chinese Academy of Sciences, Beijing 100190, PR China
**2** Marietta Blau Institute for Particle Physics (MBI), Austrian Academy of Sciences (OeAW), 1010 Vienna, Austria

September 11, 2025

## Abstract

We present a dedicated graph neural network (GNN)-based methodology for the extraction of the Higgs boson signal strength $\mu$, incorporating systematic uncertainties. The architecture features two branches: a deterministic GNN that processes kinematic variables unaffected by nuisance parameters, and an uncertainty-aware GNN that handles inputs modulated by systematic effects through gated attention-based message passing. Their outputs are fused to produce classification scores for signal-background discrimination. During training we sample nuisance-parameter configurations and aggregate the loss across them, promoting stability of the classifier under systematic shifts and effectively decorrelating its outputs from nuisance variations. The resulting binned classifier outputs are used to construct a Poisson likelihood, which enables profile likelihood scans over signal strength, with nuisance parameters profiled out via numerical optimization. We validate this framework on the FAIR Universe Higgs Uncertainty Challenge dataset, yielding accurate estimation of signal strength $\mu$ and its 68.27% confidence interval, achieving competitive coverage and interval widths in large-scale pseudo-experiments. Our code "Systematics-Aware Graph Estimator" (SAGE) is publicly available.

# 1   Introduction

With the increasing volume of data collected at the Large Hadron Collider (LHC), precision measurements are no longer limited by statistical uncertainty but by systematic effects stemming from detector calibration, theoretical modeling, and event reconstruction [1, 2]. This shift underscores the growing importance of accurately modeling multiple nuisance parameters and their combined impact on classifier outputs and likelihood inference when extracting physical observables, such as signal strengths, from experimental data. The challenge is particularly pronounced when employing machine learning classifiers for signal discrimination, as subtle systematic variations can affect the learned decision boundaries and ultimately propagate to the estimation of the signal strength and its confidence interval.

Machine learning, particularly deep neural networks, has demonstrated impressive performance in classification tasks within Higgs boson analyses [3–11]. However, most standard architectures fail to propagate or incorporate systematic uncertainties in a manner that supports downstream statistical interpretation. As a result, estimations, such as signal strengths and confidence intervals derived from likelihood functions built on classifier scores, may be biased or under-covered. This disconnect between model prediction and statistical inference undermines the robustness of likelihood-based signal strength estimation. To achieve robust and trustworthy measurements, it is therefore essential to incorporate systematic variations directly into the learning workflow, allowing the classifier to learn a smooth and statistically consistent response to these uncertainties.

Traditionally, LHC analyses simplify complex measurements by projecting high-dimensional data into low-dimensional summary statistics through binning and physically motivated observables [12–15]. This dimensional reduction is essential to make the likelihood function analytically tractable, particularly when incorporating systematic uncertainties via shape variations in histograms induced by nuisance parameters. Such binned methods remain standard in precision Higgs measurements and provide a robust framework for statistical inference. In parallel, modern machine-learning methods enable unbinned analyses by learning surrogate likelihoods or posteriors directly from high-dimensional features and nuisance-aware simulations, thereby avoiding histograming altogether. As the complexity of datasets and the number of systematic sources grow, recent developments have explored machine-learning approaches that extend beyond traditional binning. These strategies fall into several categories:

(i) **Simulation-based inference (SBI)** [16–60] directly learns an unbinned surrogate likelihood or posterior from high-dimensional features and nuisance-aware simulations, eliminating the need for binning or simplified observables. This enables precision analyses without sacrificing information from detector or modeling effects.

(ii) **Decorrelation techniques** [61–78], such as adversarial training and penalty-based regularization, seek to reduce the sensitivity of classifier outputs to nuisance parameters, enhancing robustness but without yielding an explicit likelihood function.

(iii) **Training with downstream test statistics** [79–92], integrates statistical inference goals directly into the training objective, using approximations of the profile likelihood or related test statistics to optimize model parameters in a statistically meaningful way.

(iv) **Bayesian neural networks** [93–96] aim to model uncertainties from the model and from the data by placing distributions over network weights or outputs, allowing for uncertainty estimation in predictions that can be propagated into inference procedures.

An alternative method to obtain uncertainties on neural network outputs involves ensembling together with special training objectives [97–99]. In addition to these categories, recent work [100] proposes a generative–discriminative hybrid: a contrastive normalizing-flow +

DNN pipeline trained across multiple nuisance settings to learn nuisance-aware features, with inference performed via a binned likelihood in score space.

Beyond these training paradigms, the event representation and inductive bias are equally crucial. Graph neural networks (GNNs) naturally encode collider events as variable-size sets of physics objects (nodes) linked by kinematic or geometric relations (edges); permutation-invariant message passing aggregates local neighborhoods and captures non-Euclidean, long-range correlations. Variants with dynamic edge construction and attention achieve state-of-the-art results in tracking, jet tagging, particle-flow, and event-level classification, making GNNs strong baselines for LHC analyses [101]. In this work, we present a dedicated GNN-based method for estimating the Higgs boson signal strength $\mu$ in the presence of systematic uncertainties. Our model architecture features two branches: a deterministic GNN that processes features unaffected by nuisance shifts, and an uncertainty-aware GNN that handles systematics-perturbed features. By randomly sampling 100 nuisance parameter configurations per training epoch and injecting the corresponding perturbed datasets, the classifier learns to produce outputs that are robust to systematic effects, without explicitly taking nuisance parameters as input. This framework falls into the category of decorrelation techniques, where the goal is to suppress the sensitivity of classifier outputs to nuisance variations while preserving discriminative power between signal and background. Our code "Systematics-Aware Graph Estimator" (SAGE) is publicly available [102].

We validate our method using the FAIR Universe Higgs Uncertainty Challenge [103] (FAIR-HUC) dataset, which simulates $H \rightarrow \tau\tau$ events along with major Standard Model backgrounds and multiple sources of systematic uncertainty, including six nuisance parameters: three related to calibration and three related to background normalization. Using template morphing and profile likelihood techniques, we extract $\mu$ and its confidence interval in a statistically consistent manner. The results demonstrate competitive performance in coverage and precision, while maintaining a physics-motivated interpretation of systematic effects.

The structure of this paper is as follows. In Sec. 2, we describe the simulated datasets provided in the FAIR-HUC, including available feature definitions, and the modeling of systematic uncertainties through six nuisance parameters. In Sec. 3, we present the dual-branch graph network architecture, which separates the treatment of deterministic and uncertainty-sensitive features and integrates attention and gating mechanisms in the message passing. Sec. 4 details the procedure for estimating the Higgs signal strength $\mu$ and its confidence interval using classifier outputs under systematic variations, including interpolation and numerical profiling. In Sec. 5, we validate the method using an Asimov dataset [12] and large-scale pseudo-experiments, assessing the coverage, interval width, and robustness of the inferred signal strength, as well as the accuracy and stability of the nuisance parameter estimates. We conclude in Sec. 6.

## 2 Event Samples and Nuisance Modeling

### 2.1 The $H \rightarrow \tau\tau$ process

The decay of the Higgs boson into a pair of $\tau$ leptons ($H \rightarrow \tau\tau$) plays a pivotal role in probing the Yukawa sector of the Standard Model (SM), offering direct access to the coupling of the Higgs field to third-generation leptons. Over the past decade, both the ATLAS and CMS collaborations [104–107] have accumulated extensive evidence for this process across multiple data-taking periods and center-of-mass energies, ultimately culminating in its observation at more than $5\sigma$ significance. In the era of precision Higgs physics, measurements of the $H \rightarrow \tau\tau$

rate have transitioned from inclusive cross-section determinations [108,109] to increasingly differential studies targeting distinct production modes and kinematic regions [110–112]. The $\tau\tau$ final state thus serves as a key benchmark channel for testing SM predictions and exploring Higgs boson properties at the LHC. Beyond the SM, $H \rightarrow \tau\tau$ also provides clean sensitivity to SMEFT effects: dimension-six operators can rescale or rephase the $\tau$ Yukawa and introduce CP-odd admixtures. Consequently, precision measurements of kinematic shapes in this channel place competitive constraints on the corresponding Wilson coefficients and complement global SMEFT fits [113].

Tau leptons decay promptly either leptonically ($\tau_{\text{lep}}$) into electrons or muons plus neutrinos, or hadronically ($\tau_{\text{had}}$) into narrow jets of hadrons and a neutrino. These decay channels give rise to several final-state configurations, each with distinct experimental characteristics. Fully leptonic decays yield clean signatures with reduced backgrounds but limited mass resolution due to multiple invisible neutrinos. Fully hadronic decays benefit from larger branching fractions yet face challenges from QCD jet backgrounds. The semi-leptonic topology, where one $\tau$ decays hadronically and the other leptonically, strikes a balance between signal yield and background rejection, and is the configuration adopted in FAIR-HUC.

## 2.2   Primary and derived features

This study is based on the full training dataset provided by the FAIR-HUC [103], which simulates proton-proton collisions at the LHC. The $H \rightarrow \tau_{\text{lep}}\tau_{\text{had}}$ process serves as the signal, accompanied by three dominant SM backgrounds: $Z \rightarrow \tau\tau$, top quark pair production ($t\bar{t}$), and diboson processes (primarily $WW$ with minor contributions from $WZ$ and $ZZ$). Both the signal process and the three SM backgrounds are generated using PYTHIA8.2 [114] for parton-level event generation, followed by fast detector simulation with DELPHES3.5.0 [115]. The dataset includes only events with exactly one electron or muon, exactly one hadronically decaying $\tau$, up to two reconstructed jets, and missing transverse energy. Basic kinematic selections are applied at the generator level: the lepton is required to have transverse momentum $p_T^{\ell} > 20$ GeV and pseudorapidity $|\eta^{\ell}| < 2.5$, while the hadronic $\tau$ must satisfy $p_T^{\tau_{\text{had}}} > 26$ GeV and $|\eta^{\tau_{\text{had}}}| < 2.69$. In order to maximize statistical power and avoid selection-induced biases in classifier training, we do not perform event-level selections and retain all available training events.

The original FAIR-HUC dataset provides 28 features for each event, designed to capture the essential kinematic information of the $\tau_{\text{had}}$, the lepton, reconstructed jets, and missing transverse momentum (MET). These include primary observables such as transverse momentum ($p_T$), pseudorapidity ($\eta$), and azimuthal angle ($\phi$) of the visible final-state particles, as well as the missing transverse momentum and its azimuthal angle. Additional variables summarize global properties of the event, such as the number of reconstructed jets, the scalar sum of all the jets' $p_T$ of the events. The derived features encode correlations between particles and characterize the overall event topology. These include the transverse mass of the lepton and MET ($m_T$), the visible invariant mass of the lepton and $\tau_{\text{had}}$ system ($m_{\text{vis}}$), and the modulus of the vector sum of the lepton, $\tau_{\text{had}}$, and MET momenta ($p_T^{\text{H}}$). Additional features include the invariant mass of the two leading jets, the ratio of lepton to $\tau_{\text{had}}$ transverse momentum, and the angular separation $\Delta R$ between the lepton and $\tau_{\text{had}}$. The global event kinematics is characterized by the scalar and vector sums of all reconstructed transverse momenta ($\sum p_T$ and $p_T^{\text{tot}}$), while centrality and topology are captured by the azimuthal centrality of MET with respect to the lepton and $\tau_{\text{had}}$ ($C_\phi^{\text{miss}}$), and the pseudorapidity centrality of the lepton with respect to the jets ($C_\eta^{\ell}$). Finally, the pseudorapidity separation and the product of pseudorapidities of the two leading jets are included as jet-correlated shape observables. These engineered features are designed to enhance sensitivity to the Higgs boson decay topology and improve background rejection. Participants in the FAIR-HUC are free to utilize, modify, or omit

Table 1: List of event features used in the analysis. Primary (PRI) variables are direct kinematic observables, while derived (DER) variables encode higher–level correlations and global event properties. Features shown in blue denote the additional nine angular distance features introduced in this study.

| Primary Features (PRI) | | | |
|---|---|---|---|
| **Symbol** | **Description** | **Symbol** | **Description** |
| $p_T^\ell$ | Transverse momentum of the lepton | $\eta^\ell$ | Pseudorapidity of the lepton |
| $\phi^\ell$ | Azimuthal angle of the lepton | $p_T^{\tau_{had}}$ | Transverse momentum of the hadronic $\tau$ |
| $\eta^{\tau_{had}}$ | Pseudorapidity of the hadronic $\tau$ | $\phi^{\tau_{had}}$ | Azimuthal angle of the hadronic $\tau$ |
| $p_T^{j_1}$ | Transverse momentum of the leading jet | $\eta^{j_1}$ | Pseudorapidity of the leading jet |
| $\phi^{j_1}$ | Azimuthal angle of the leading jet | $p_T^{j_2}$ | Transverse momentum of the subleading jet |
| $\eta^{j_2}$ | Pseudorapidity of the subleading jet | $\phi^{j_2}$ | Azimuthal angle of the subleading jet |
| $N_j$ | Number of reconstructed jets | $\sum_{\text{jets}} p_T$ | Scalar sum of transverse momenta of all jets |
| $\vec{p}_T^{\text{miss}}$ | Missing transverse momentum | $\phi^{\text{miss}}$ | Azimuthal angle of missing transverse momentum |
| **Derived Features (DER)** | | | |
| **Symbol** | **Description** | **Symbol** | **Description** |
| $m_T(\ell, \vec{p}_T^{\text{miss}})$ | Transverse mass of lepton and $\vec{p}_T^{\text{miss}}$ | $m_{\text{vis}}$ | Visible invariant mass of $\tau_{had}$ and $\ell$ |
| $p_T^H$ | Vector sum of $p_T^{\tau_{had}}, p_T^\ell, \vec{p}_T^{\text{miss}}$ | $m^{j_1 j_2}$ | Invariant mass of the two leading jets |
| $\Delta R(\ell, miss)$ | Angular distance between $\ell$ and MET | $\Delta R(\tau_{had}, miss)$ | Angular distance between $\tau_{had}$ and MET |
| $\Delta R(j_1, miss)$ | Angular distance between $j_1$ and MET | $\Delta R(j_2, miss)$ | Angular distance between $j_2$ and MET |
| $\Delta R(j_1, \tau_{had})$ | Angular distance between $j_1$ and $\tau_{had}$ | $\Delta R(j_2, \tau_{had})$ | Angular distance between $j_2$ and $\tau_{had}$ |
| $\Delta R(j_1, \ell)$ | Angular distance between $j_1$ and $\ell$ | $\Delta R(j_2, \ell)$ | Angular distance between $j_2$ and $\ell$ |
| $\Delta R(j_1, j_2)$ | Angular distance between $j_1$ and $j_2$ | $\Delta R(\tau_{had}, \ell)$ | Angular distance between $\tau_{had}$ and $\ell$ |
| $p_T^{\text{tot}}$ | Vector sum of all visible momenta and $\vec{p}_T^{\text{miss}}$ | $\sum p_T$ | Scalar sum of all visible momenta and $\vec{p}_T^{\text{miss}}$ |
| $C_\phi^{\text{miss}}$ | Azimuthal centrality of $\vec{p}_T^{\text{miss}}$ w.r.t. $\ell, \tau_{had}$ | $C_\eta^\ell$ | Pseudorapidity centrality of the lepton w.r.t. the jets |
| $p_T^\ell / p_T^{\tau_{had}}$ | Transverse–momentum ratio of lepton to $\tau_{had}$ | | |

these features depending on their analysis strategy. In our implementation, we exclude two variables—pseudorapidity separation and product of pseudorapidities of the leading jets—and extend the original feature set with nine additional angular distance variables between all pairs of final-state particles. This modification is tailored to the architecture of our graph neural network, which benefits from explicitly encoding geometric relations between nodes. A complete list of the 35 features used in our model is summarized in Table 1, while the definitions of the original 28 FAIR-HUC observables are available in Ref. [103].

## 2.3 Systematic uncertainties

To reflect realistic sources of experimental uncertainty, each event in the FAIR-HUC dataset is subject to systematic variations governed by six predefined nuisance parameters, grouped into two categories:

- **Calibration-type uncertainties** affect the primary features of final-state objects directly and propagate to derived features. These include the jet energy scale (JES), $\tau$ energy scale (TES), and the soft missing transverse energy (soft MET) scale. The JES and TES variations are parameterized by standard normal nuisance parameters, $\alpha_{\text{jes}}, \alpha_{\text{tes}} \sim \mathcal{N}(1, 0.01)$, subject to the constraint $0.9 \leq \alpha_{\text{tes}}, \alpha_{\text{jes}} \leq 1.1$, ensuring realistic perturbations. The MET scale variation is controlled by a log-normal nuisance parameter $\alpha_{\text{met}}$ with mean 0 and unit width in log-space, constrained to the range $[0, 5]$. For all three sources, the FAIR-HUC organizers

provide code to modify both primary and derived features as a function of these nuisance parameters.

- **Normalization-type uncertainties** affect only the event weights of background processes. These include a global background normalization factor ($\alpha_{\text{bkg}}$) and process-specific modifiers for the $t\bar{t}$ ($\alpha_{t\bar{t}}$) and diboson ($\alpha_{\text{VV}}$) processes. All three nuisance parameters follow Gaussian distributions centered at 1, with standard deviations $\sigma_{\text{bkg}} = 0.001$, $\sigma_{t\bar{t}} = 0.02$, and $\sigma_{\text{VV}} = 0.25$, and are clipped to physically motivated ranges $[0.99, 1.01]$, $[0.8, 1.2]$ and $[0, 2]$, respectively. Their effect is implemented via rescaling of the event weights:

$$w^{\text{biased}}_{Z \to \tau\tau} = \alpha_{\text{bkg}} \times w_{Z \to \tau\tau}, \quad w^{\text{biased}}_{t\bar{t}} = \alpha_{\text{bkg}} \times \alpha_{t\bar{t}} \times w_{t\bar{t}}, \quad w^{\text{biased}}_{\text{VV}} = \alpha_{\text{bkg}} \times \alpha_{\text{VV}} \times w_{\text{VV}}.$$

These transformations preserve the kinematic shapes of the background processes while altering their relative and total contributions, following the parameter definitions from the FAIR-HUC [103].

Together, the six nuisance parameters account for dominant sources of systematic uncertainty in both event kinematics and background composition. Their effect is consistently propagated across both primary and derived features. This design mirrors realistic experimental challenges, where the true values of nuisance parameters are unknown, and robust performance under systematic shifts is essential for reliable statistical inference. The impact of these uncertainties is quantified downstream in the signal strength fit, as discussed in Sec.4 and validated via Monte Carlo studies in Sec.5.

# 3 Dual-Branch Graph Network Architecture

## 3.1 Overall Model Architecture and Workflow

To enable robust signal strength estimation under systematic uncertainties, we develop a graph-based classification model that processes each event using a dual-branch architecture. The model takes the full set of 35 features as input. These features capture the essential information of final-state particles and event topology, and are structured into a physics-motivated graph representation. We adopt a GNN for three practical reasons. First, collider events are intrinsically relational: discrimination depends on both pairwise correlations (e.g., $\Delta R$, invariant masses) and higher-order (three-object or more) structures—e.g., the azimuthal centrality of $p_T^{\text{miss}}$ with respect to $(\ell, \tau_{\text{had}})$—which message passing captures naturally. Second, the graph representation is permutation-invariant—message passing with symmetric (sum/mean) aggregation makes the prediction invariant to the arbitrary indexing of nodes within the event graph—and tolerant to missing objects via masking, providing robustness to varying jet multiplicities and threshold effects without ad-hoc ordering. Third, the node/edge/global split offers clean attachment points for systematics: calibration shifts on single objects or object pairs map naturally to nodes/edges, whereas calibration effects that couple three or more objects—handled here as "global" uncertainties—are injected as event-level context, and their influence propagates in a controlled, physically interpretable way.

By decoupling the processing of deterministic and uncertainty-sensitive features, the dual-branch design allows the model to leverage both sources of information for signal-background discrimination, while also ensuring a smooth and robust dependence of its predictions on nuisance parameter variations. A dedicated fusion module combines the learned representations from both branches, and the model is trained on systematically perturbed samples to ensure stable and controlled classifier responses in the downstream likelihood profiling. In the following, we describe the input representation in Section 3.2, the deterministic branch in

Table 2: Event statistics of the FAIR-HUC training dataset. The raw event counts are the total numbers of simulated events provided in the dataset, which are subsequently split 90/10 into training and validation subsets. The event yields are the sums of per-event physical weights, *i.e.* their expected number based on process cross section and luminosity.

| Process | Raw event count | Event yield |
|---|---|---|
| $H \to \tau\tau$ | 49450685 | 966.0 |
| $Z \to \tau\tau$ | 90113695 | 901137.5 |
| $t\bar{t}$ | 4128325 | 41283.4 |
| VV | 343294 | 3433.5 |

Section 3.3, the uncertainty-aware branch in Section 3.4, and the fusion module in Section 3.5.

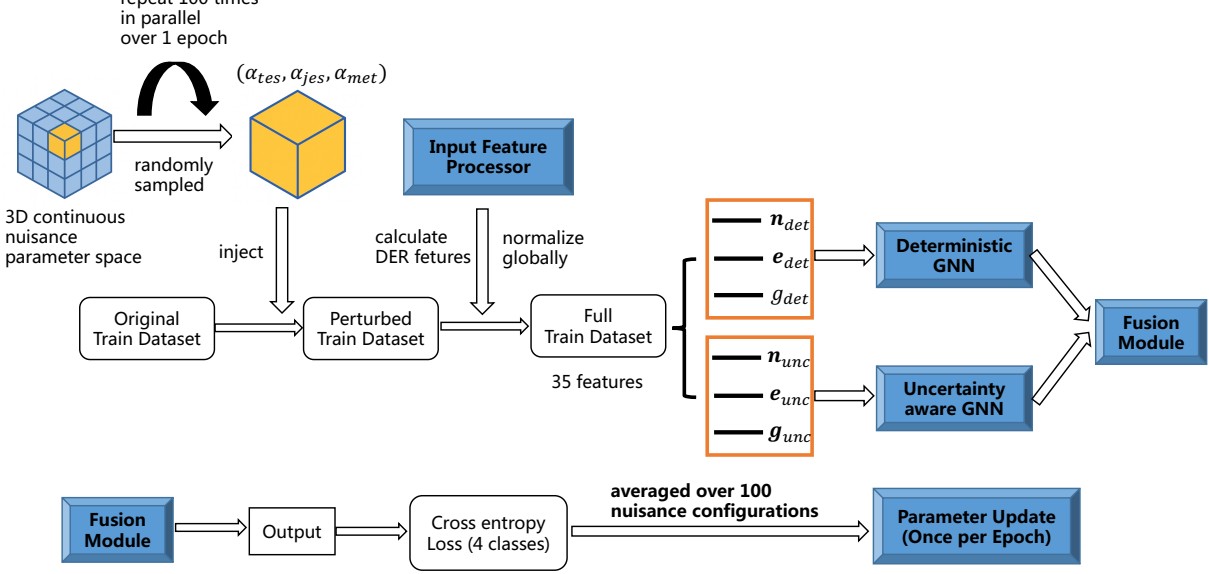

Figure 1: Illustration of the overall model architecture and training strategy.

The training procedure, illustrated in Figure 1, is designed to incorporate systematic uncertainties in a physically meaningful and computationally efficient manner. The full training set provided by FAIR-HUC is used without any event selection, and a randomly selected 10% subset is reserved for validation. Classification is performed with uniform event weighting, i.e., every event is assigned the same weight irrespective of its class. The raw and weighted event counts for all processes are summarized in Table 2, where it is noted that the event counts are not based on luminosity times cross section and are not balanced across classes. Importantly, although the total dataset includes six nuisance parameters (three calibration-based and three normalization-based), only the three calibration-type parameters—the $\tau$ energy scale (TES), the jet energy scale (JES) and the soft MET scale—are used in the perturbation procedures. This choice reflects the fact that the training data itself does not contain normalization weights. The training dataset has a shape of $(N, 17)$, consisting of 16 primary features and one label. All 19 derived features (DER) used in the graph model are computed entirely on the fly, immediately after applying nuisance perturbations to the relevant primary (PRI) features, rather than being precomputed.

The training strategy incorporates stochastic sampling over nuisance parameters to marginal-

ize systematic effects during training. At the beginning of each epoch, 100 distinct nuisance parameter vectors are sampled from their respective prior distributions (see Section 3.4 for details). For each training batch, the nominal events are duplicated $R = 100$ times, called replicas, with each replica perturbed by one of the sampled nuisance configurations. The deterministic GNN branch processes the nominal features only once, while the uncertainty-aware branch simultaneously and independently processes all 100 perturbed variants in parallel. The two branches are merged via a gated fusion module, and the final prediction is obtained using a four-class cross-entropy loss, enabling the model to separate the signal from each of the three major SM background classes and to discriminate among the background classes. Denoting the model output for replica $r$ of event $i$ as $p^{(i,r)}$ and the one-hot target as $y^{(i)}$, the training objective is

$$\mathcal{L} = \frac{1}{BR} \sum_{i=1}^{B} \sum_{r=1}^{R} \left( - \sum_{c=1}^{4} y_c^{(i)} \log p_c^{(i,r)} \right), \tag{1}$$

where $B$ is the batch size and $c$ indicates the signal/background class. For each training batch, the loss is computed for all 100 perturbed replicas of every event, and a single scalar loss value is obtained by averaging over all replicas and events in the batch. A single backward pass and parameter update is then performed using this averaged loss. This strategy encourages the model to produce predictions that are insensitive to nuisance-induced variations, while preserving discriminative power between signal and background across the relevant phase space.

The model is optimized using the AdamW algorithm [116], with a learning rate governed by a cosine annealing schedule characterized by a maximum iteration count of $T_{\max} = 20$, a minimum learning rate of $\eta_{\min} = 10^{-5}$, and a weight decay factor of $10^{-4}$. Training is carried out for 12 epochs, requiring approximately two days on a single NVIDIA Tesla V100 GPU, largely driven by the 100 times $3 \cdot 10^8$ events dataset size. For validation, at the start of each epoch we draw a single nuisance-parameter vector at random and evaluate the loss once at that fixed configuration. Overall, the training strategy emphasizes robustness by integrating out systematic effects through stochastic averaging, supporting the construction of a robust model with *reduced and controlled sensitivity* to nuisance fluctuations—rather than full decorrelation— and suitable for downstream statistical inference tasks such as maximum likelihood estimation and coverage evaluation.

To connect the trained classifier with the statistical inference pipeline, we introduce an inclusive region for extracting the signal strength and three control regions for constraining background normalization nuisance parameters, with precise definitions given in Sec. 4. The downstream goal of the GNN is to turn the signal-class output probability $p_{H \to \tau\tau}$ into a binned observable per analysis region and then perform likelihood inference. Concretely, for each region we histogram $p_{H \to \tau\tau}$ using region-specific adaptive binning. Using simulated predictions at fixed calibration shifts, we precompute a region-wise interpolation table $T_R(\alpha_{\text{tes}}, \alpha_{\text{jes}}, \alpha_{\text{met}})$ that maps any nuisance configuration to process-wise binned yields $\{s_{b,R}, z_{b,R}, t_{b,R}, v_{b,R}\}$. The surrogate likelihood is then formed as a product of Poisson terms over all bins: in the inclusive region the expectation includes the signal scaled by $\mu$ plus all background components, whereas in the control regions only backgrounds enter. We profile the nuisances to obtain $\hat{\mu}$ and extract confidence intervals from the profiled NLL curve. Full details of the downstream pipeline (adaptive binning, interpolation tables, likelihood construction, and profiling) are given in Section 4.

## 3.2   Input Representation

To enable efficient learning from collider events, we transform each event into a graph-structured representation that captures both the properties of individual particles and their kinematic correlations. The input representation consists of three components: node features $\mathbf{n}$, edge features $\mathbf{e}$, and global features $\mathbf{g}$, encompassing a total of 35 features. These features comprise both deterministic quantities and those affected by systematic variations. To account for their distinct characteristics, the two types are processed separately within the architecture. While the present study focuses on $H \to \tau\tau$ final states, the graph-based representation is particularly advantageous for more complex topologies and higher object multiplicities, where permutations of identical particles and correlations among them become more relevant.

Each event is encoded as a graph with five nodes, corresponding to the final-state objects: lepton, tau-jet ($\tau_{\mathrm{had}}$), leading jet ($j_1$), subleading jet ($j_2$), and missing transverse momentum (MET). For each node, we construct two types of features: deterministic and uncertainty-aware. The deterministic node features include the lepton transverse momentum $p_T$ together with pseudorapidity $\eta$ and azimuthal angle $\phi$ of all the visible final objects, while uncertainty-aware node features consist of quantities sensitive to the three calibration-type nuisance parameters. These include the transverse momentum of the $\tau$ jet (affected by $\alpha_{\mathrm{tes}}$), the transverse momentum of the two QCD jets (affected by $\alpha_{\mathrm{jes}}$), and both the transverse momentum and azimuthal angle of the MET (affected by $\alpha_{\mathrm{met}}$). For each node, the coordinates that do not belong to its current branch are set to zero. For example, the deterministic GNN branch zero-pads the tau-jet $p_T$ slot, whereas the uncertainty-aware branch zero-pads the tau-jet $\eta$ and $\phi$ slots. And each node is additionally tagged with a five-dimensional one-hot vector to indicate its identity. Table 3 summarizes the structure and assignments of node features $\mathbf{n}$.

To characterize the spatial and kinematic relationships between final-state objects, we define edge features for each valid pair of nodes. These include angular separations as well as physics-motivated variables such as the invariant mass $m_{j_1 j_2}$ (when both jets are present) and the transverse mass $m_{\mathrm{T}}(\ell, \vec{p}_{\mathrm{T}}^{\mathrm{miss}})$. Edge features are classified as deterministic or uncertainty aware depending on whether they are affected by systematic variations; for example, $m_{j_1 j_2}$ depends on the jet energy scale, while $\Delta R(j_1, j_2)$ is deterministic. If one or both particles in a pair are missing, the corresponding edge is simply omitted. Table 4 provides a complete list of all edge features $\mathbf{e}$ and their assignments.

In addition to node and edge features, we extract seven global features $\mathbf{g}$ constructed from three or more final-state objects, such as the vector sum of all visible momenta and $\vec{p}_{\mathrm{T}}^{\mathrm{miss}}$ or the pseudo rapidity centrality of the lepton with respect to the jets. These features are partitioned into deterministic and uncertainty-aware subsets according to their dependence on systematically affected quantities. The definitions of global features $\mathbf{g}$ are given in Table 5.

To ensure numerical stability and consistency across nominal and systematically perturbed inputs, all features are standardized using global mean and standard deviation statistics computed once on the training set. These statistics are then frozen and reused throughout: for both the nominal and systematics-perturbed datasets, during both training and validation. Each feature dimension is normalized independently (channel-wise) using its mean and standard deviation of the nominal training dataset. For derived features that depend on jet availability, we construct three variants of the feature matrix (both jets present, leading-only, or neither). Each variant includes only the computable quantities, which are then standardized using the same global training-set statistics (not variant-specific ones). Based on these standardized tensors, we form the deterministic and uncertainty-sensitive node/edge/global inputs that feed the two GNN branches.

Table 3: Node feature assignment for final-state objects. Each node is represented by a one-hot encoding indicating its type. Deterministic and uncertainty-aware features are assigned separately according to their sensitivity to systematic variations.

| Node Type | Feature Type | Variables Used | Feature Role | Node Encoding |
|---|---|---|---|---|
| **Lepton** | Deterministic | $p_T, \eta, \phi$ | Fully specified | [1, 0, 0, 0, 0] |
| **Tau-jet** | Deterministic | $\eta, \phi$ | Spatial info | [0, 1, 0, 0, 0] |
| | Uncertainty-aware | $p_T$ | Affected by $\alpha_{\text{tes}}$ | [0, 1, 0, 0, 0] |
| **Leading Jet** | Deterministic | $\eta, \phi$ | Spatial info | [0, 0, 1, 0, 0] |
| | Uncertainty-aware | $p_T$ | Affected by $\alpha_{\text{jes}}$ | [0, 0, 1, 0, 0] |
| **Subleading Jet** | Deterministic | $\eta, \phi$ | Spatial info | [0, 0, 0, 1, 0] |
| | Uncertainty-aware | $p_T$ | Affected by $\alpha_{\text{jes}}$ | [0, 0, 0, 1, 0] |
| **MET** | Uncertainty-aware | $p_T, \phi$ | Affected by $\alpha_{\text{met}}$ | [0, 0, 0, 0, 1] |

Table 4: Edge features constructed between pairs of final-state objects. Features are categorized as deterministic or uncertainty-aware based on their sensitivity to systematic variations.

| Edge Type | Feature Name | Feature Type |
|---|---|---|
| Tau-jet – MET | $\Delta R(\tau_{\text{had}}, \text{miss})$ | Uncertainty-aware |
| Lepton – MET | $\Delta R(\ell, \text{miss})$ | Uncertainty-aware |
| | $m_{\text{T}}(\ell, p_{\text{T}}^{\text{miss}})$ | Uncertainty-aware |
| Jet1 – MET | $\Delta R(j_1, \text{miss})$ | Uncertainty-aware |
| Jet2 – MET | $\Delta R(j_2, \text{miss})$ | Uncertainty-aware |
| Tau-jet – Jet1 | $\Delta R(\tau_{\text{had}}, j_1)$ | Deterministic |
| Tau-jet – Jet2 | $\Delta R(\tau_{\text{had}}, j_2)$ | Deterministic |
| Lepton – Jet1 | $\Delta R(\ell, j_1)$ | Deterministic |
| Lepton – Jet2 | $\Delta R(\ell, j_2)$ | Deterministic |
| Jet1 – Jet2 | $\Delta R(j_1, j_2)$ | Deterministic |
| | $m^{j_1 j_2}$ | Uncertainty-aware |
| Tau-jet – Lepton | $\Delta R(\tau_{\text{had}}, \ell)$ | Deterministic |
| | $p_{\text{T}}^{\ell}/p_{\text{T}}^{\tau_{\text{had}}}$ | Uncertainty-aware |
| | $m_{\text{vis}}(\tau_{\text{had}}, \ell)$ | Uncertainty-aware |

Table 5: Global features constructed from three or more final-state objects. Each feature is categorized by its defining particles and its sensitivity to systematic uncertainties.

| Particles Involved | Feature Name | Feature Type |
|---|---|---|
| All QCD jets | $N_j$ | Uncertainty-aware |
| Tau-jet, Lepton, Jet1, Jet2, MET | $p_{\mathrm{T}}^{\mathrm{tot}}$ | Uncertainty-aware |
| Tau-jet, Lepton, MET | $p_{\mathrm{T}}^H$ | Uncertainty-aware |
| Tau-jet, Lepton, Jet1, Jet2, MET | $\sum p_{\mathrm{T}}$ | Uncertainty-aware |
| Tau-jet, Lepton, MET | $C_\phi^{\mathrm{miss}}$ | Uncertainty-aware |
| All QCD jets | $\sum_{\mathrm{jets}} p_{\mathrm{T}}$ | Uncertainty-aware |
| Lepton, Jet1, Jet2 | $C_\eta^\ell$ | Deterministic |

## 3.3  Deterministic GNN Branch

The deterministic branch is designed to process features that are unaffected by systematic variations. It follows a graph-based message passing architecture, where encoded node features interact via predefined edge connections, and are further modulated by global event-level information. The architecture of the deterministic GNN branch is illustrated in Figure 2. The input to this branch comprises three components: deterministic node features $\mathbf{n_{det}}$, deterministic edge features $\mathbf{e_{det}}$, and a single deterministic global feature $g_{det}$. Each event graph contains five nodes, and the node features are structured as vectors of dimension 8, consisting of three physical quantities ($p_T$, $\eta$, $\phi$) and a five-dimensional one-hot encoding that identifies the particle type. When flattened across the batch, the input node tensor has shape $(20480, 8)$, corresponding to 4096 events with five nodes each. Among the 15 physical quantities, only nine deterministic features are assigned non-zero values, while all other entries are intentionally zero-padded to reserve fixed positions for uncertainty-aware features. In addition to the node features, the input includes six deterministic edge features and one deterministic global feature.

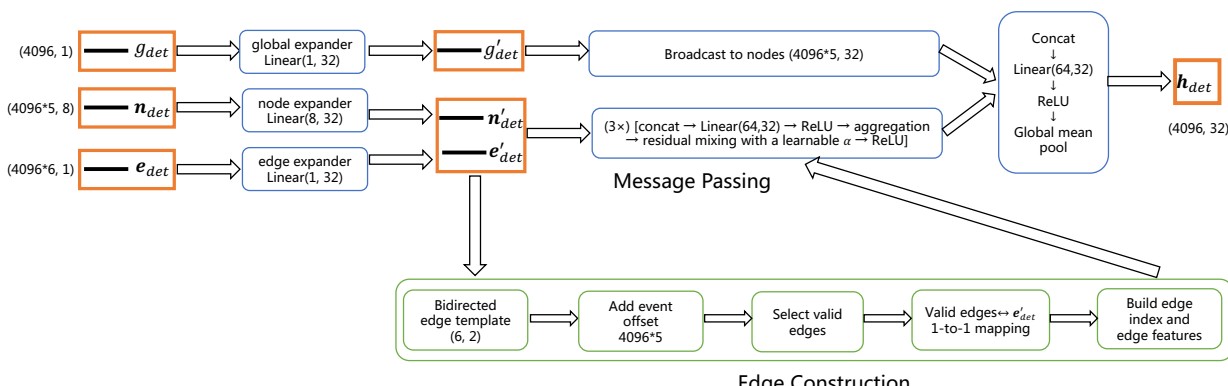

Figure 2: Illustration of the deterministic GNN architecture.

The 8-dimensional node vectors are passed through a linear node encoder to obtain per-node hidden representations. In parallel, each of the six deterministic edge features is expanded via a linear projection to a shared embedding space and assigned to the corresponding bidirected edge, as defined by a fixed edge template $\mathcal{T} \in \mathbb{Z}^{6\times2}$ that is consistently applied across all events. To accommodate batched processing, the same template is applied across all events, and event-wise node offsets, multiples of five corresponding to the number of nodes per graph,

are added to correctly index nodes in the flattened batch. The graph construction module builds the edge index and attributes for each event based on a set of valid bidirected edges, conditioned on the presence or absence of the leading and subleading jets. Invalid edges involving missing nodes are automatically excluded.

The encoded node and edge features are processed through three message passing layers. In each layer $\ell = 1, 2, 3$, messages from sender node $i$ to receiver node $j$ are computed as:

$$\mathbf{m}_{i \to j}^{(\ell)} = \text{ReLU}\left(\mathbf{W}_\ell^{\text{msg}}\left[\mathbf{h}_i^{(\ell-1)} \,\|\, \mathbf{e}_{ij}\right] + \mathbf{b}_\ell^{\text{msg}}\right), \tag{2}$$

where $\|$ denotes vector concatenation; $\mathbf{h}_i^{(\ell-1)}$ is the node-$i$ embedding from the previous layer, $\mathbf{e}_{ij}$ is the encoded edge feature for node pair $(i, j)$, $\mathbf{W}_\ell^{\text{msg}}$ and $\mathbf{b}_\ell^{\text{msg}}$ are learnable parameters at layer $\ell$. At each node, the messages from its connected nodes are summed together:

$$\mathbf{a}_j^{(\ell)} = \sum_{i \in \mathcal{N}(j)} \mathbf{m}_{i \to j}^{(\ell)}, \tag{3}$$

and combined with the previous node states via residual mixing

$$\mathbf{h}_j^{(\ell)} = \text{ReLU}\left(\alpha \cdot \mathbf{h}_j^{(\ell-1)} + (1-\alpha) \cdot \mathbf{a}_j^{(\ell)}\right), \tag{4}$$

where $\alpha$ is a learnable scalar gate, shared across nodes and layers, and hard-projected to $[0, 1]$ at each forward pass.

To incorporate global context, the global deterministic feature is passed through a separate linear encoder and broadcasted to all five nodes per graph. The node states are updated by concatenating them with the broadcasted global embedding and applying another linear transformation followed by ReLU activation. Finally, the updated node features are aggregated across all nodes within each event by taking their element-wise mean, a procedure known as global mean pooling. This operation converts a variable-size set of node representations into a single fixed-size graph-level embedding, resulting in a tensor of shape $(4096, 32)$ per batch. This vector, $\mathbf{h}_{\text{det}}$, serves as the deterministic summary of each event, capturing spatial, kinematic, and topological correlations among visible particles through message passing.

## 3.4 Uncertainty-aware GNN Branch

The uncertainty-aware branch is designed to process node features $\mathbf{n}_{\text{unc}}$, edge features $\mathbf{e}_{\text{unc}}$, and global features $\mathbf{g}_{\text{unc}}$ that are sensitive to systematic variations. Conceptually, this branch mirrors the deterministic one in its graph-based message passing structure, but incorporates additional attention and gating mechanisms that enable adaptive modulation of the information flow. This design is introduced only in the uncertainty-aware branch, because the deterministic inputs are fixed quantities, whereas the uncertainty-aware inputs explicitly encode features that vary under systematic perturbations and therefore require adaptive weighting. This design is consistent with our training objective, which averages the cross-entropy over $R=100$ nuisance replicas per event and thus averages out fluctuations in the predictions under systematic shifts. The architecture of the uncertainty-aware GNN branch is illustrated in Figure 3. The input to this branch consists of three components: uncertainty-aware node features of shape $(20480, 8)$, uncertainty-aware edge features of shape $(32768, 1)$, corresponding to eight bidirected edges per event across a batch of 4096 events, and global uncertainty-aware features of shape $(4096, 6)$, representing six global features per event. Each event graph contains five nodes, and among the 15 physical quantities assigned to nodes, only five uncertainty-aware variables are given non-zero values while all other entries are zero-padded to maintain alignment with the deterministic input structure. Edge construction reuses the same logic as the deterministic

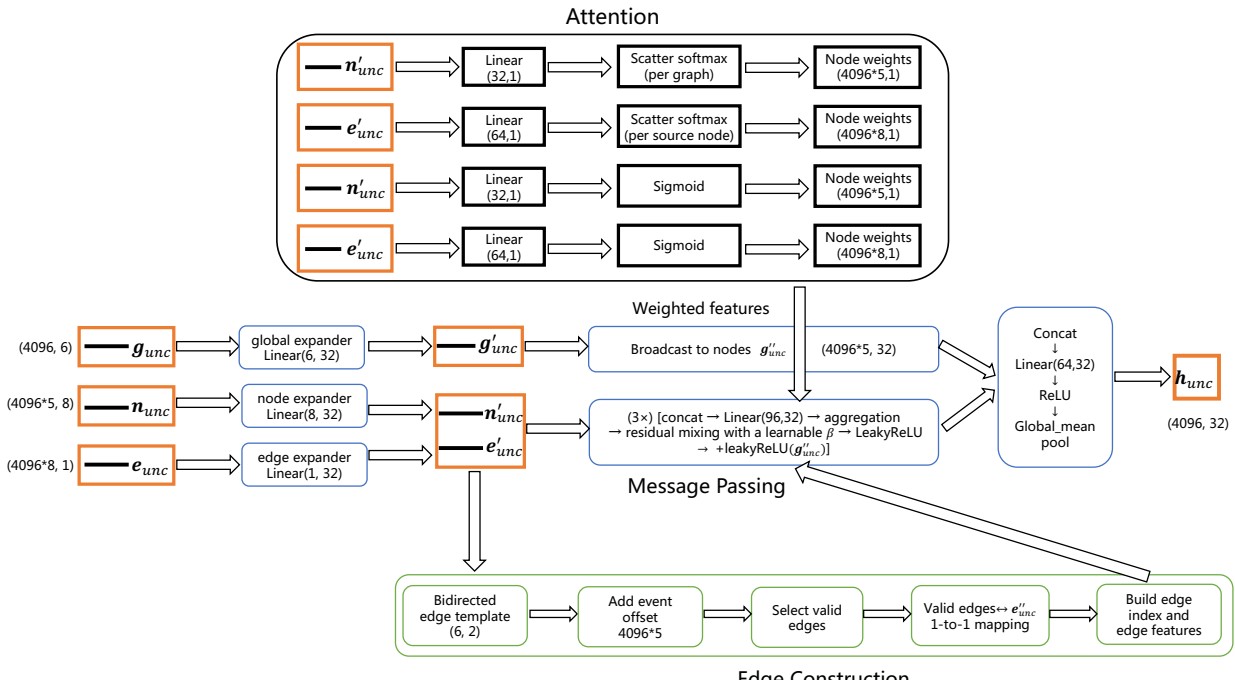

Figure 3: Illustration of the uncertainty-aware GNN architecture.

branch, with masking based on jet availability to select valid edges and index offsets applied to correctly locate nodes in the flattened batch.

At the beginning of forward propagation, the 8-dimensional node feature vectors are passed through a linear encoder followed by LeakyReLU activation, producing the initial node embeddings. These embeddings are then used to compute scalar attention scores via a linear layer of shape $(32, 1)$, followed by softmax normalization within each event. In parallel, a separate linear layer of shape $(32, 1)$ with sigmoid activation is applied to each node embedding to produce node-specific gates that modulate the contribution of self-information during aggregation. For edges, each of the eight scalar edge features is first projected into a shared 32-dimensional embedding space via a linear transformation with LeakyReLU activation. Meanwhile, the 6-dimensional global feature vector for each event is passed through a linear encoder into a 32-dimensional representation, which is then broadcasted to all five nodes in the corresponding event graph. For each edge, its embedding is concatenated with the corresponding global context vector associated with the sender node, forming a 64-dimensional input. This is fed into two separate linear layers: one of shape $(64, 1)$ and softmax normalization to compute edge attention weights, and another with shape of $(64, 1)$ and sigmoid activation to compute learnable edge gates that modulate the information flow during message passing.

To perform uncertainty-aware message passing, the model uses the previously computed attention and gating weights to construct and propagate messages across the event graph. For each edge $(i \rightarrow j)$, the modulated sender node representation and edge embedding are computed as:

$$h_i^{\text{mod}} = a_i \cdot g_i \cdot h_i \tag{5}$$

$$e_{ij}^{\text{mod}} = a_{ij} \cdot g_{ij} \cdot e_{ij} \tag{6}$$

where $a_i$ and $a_{ij}$ denote the attention weights for node $i$ and edge $(i, j)$, respectively, and $g_i$ and $g_{ij}$ represent the corresponding gating weights. The message is then formed by concatenating

$h_i^{\mathrm{mod}}$, $e_{ij}^{\mathrm{mod}}$, and the global context $g_e$ of the corresponding event (broadcasted to all nodes):

$$m_{ij} = \phi\left(\mathrm{concat}(h_i^{\mathrm{mod}}, e_{ij}^{\mathrm{mod}}, g_e)\right) \tag{7}$$

where $\phi$ denotes a linear transformation of shape $(96, 32)$ followed by LeakyReLU activation, which outputs learned message vectors. These modulated messages are then aggregated at the receiver nodes using a scatter-based summation operation over all incoming edges:

$$\hat{h}_j = \sum_{i \in \mathcal{N}(j)} m_{ij} \tag{8}$$

To preserve existing node representations and stabilize training, the aggregated messages are combined with the previous node embeddings via residual mixing:

$$h_j^{\mathrm{new}} = \mathrm{LeakyReLU}\left(\beta h_j + (1 - \beta)\hat{h}_j\right) \tag{9}$$

where $\beta \in [0, 1]$ is a learnable scalar parameter shared across the model and governs the balance between the new incoming messages and the previous hidden states. After the residual update, a LeakyReLU activation is applied. Additionally, the global feature embedding is added to the node representation at every layer to reinforce global event-level information.

The entire message passing process is repeated three times with independent edge encoders but a shared attention matrix, which enforces a consistent notion of importance across layers while still allowing the model to capture higher-order interactions among the final-state particles. After the final propagation step, the updated node representations are concatenated with their corresponding broadcasted global feature vectors and passed through a final linear transformation $(64, 32)$ to produce the final node states. To generate a per-event summary representation, global mean pooling is applied across the five nodes within each event. This yields a final uncertainty-aware embedding, $\mathbf{h_{unc}}$, of shape $(4096, 32)$, which encapsulates the structural, kinematic, and topological characteristics of each event, modulated by the underlying systematic variations.

## 3.5 Fusion Module

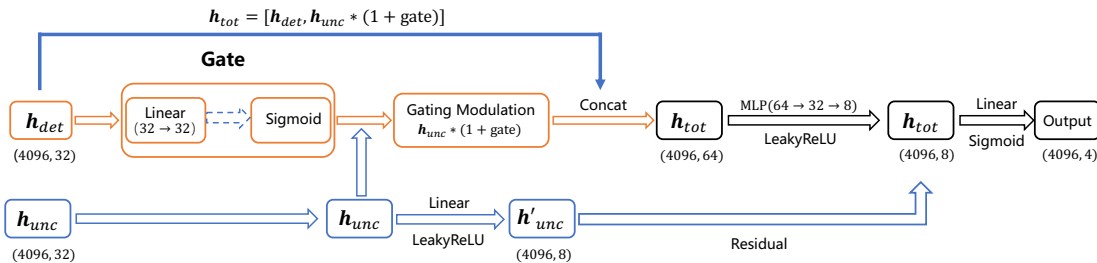

Figure 4: Illustration of the final fusion module.

The fusion module combines the output embeddings produced by the deterministic $\mathbf{h_{det}}$ and uncertainty-aware $\mathbf{h_{unc}}$ branches and transforms the joint representation into the final prediction, as illustrated in Figure 4. Each branch outputs a 32-dimensional feature vector for every event in the batch. These two vectors are first processed by a gating mechanism that adaptively modulates the uncertainty-aware embedding using information from the deterministic representation. This design is motivated by the observation that deterministic features are typically more stable and reliable, making them well-suited to guide the modulation of

uncertainty-sensitive information in a data-driven and physically informed manner. Specifically, the deterministic feature vector is passed through a linear layer of shape (32, 32), followed by a sigmoid activation, producing a gate vector with shape (4096, 32) in the range of [0, 1]. This gate acts as a set of soft, learnable scaling factors applied element-wise to the uncertainty-aware embedding, which is then shifted by adding 1 to each component—thereby enhancing it according to the deterministic context.

$$\mathbf{h}_{\text{unc\_enhanced}} \Rightarrow \mathbf{h}_{\text{unc}} \times (1 + \text{gate}) \tag{10}$$

This operation enhances each component of the uncertainty-aware vector in a way that depends on the corresponding deterministic information.

These transformations allow the model to capture nonlinear interactions between the two branches and extract higher-level features for the final prediction. To preserve direct information from the uncertainty-aware branch, a residual connection is introduced. The original 32-dimensional uncertainty-aware vector is projected to 8 dimensions through a separate linear layer with LeakyReLU, and added element-wise to the 8-dimensional output of the main path, yielding the final hidden representation. Finally, the resulting 8-dimensional vector is passed through a final linear layer to produce the model output, a 4-dimensional vector for multi-class classification. The output corresponds to raw logits; during training the softmax operation is implicitly applied within the cross-entropy loss, while during inference an explicit softmax function can be added to obtain class probabilities.

## 3.6 Signal-Class Output Distributions under Nuisance Variations

To evaluate the four-class classifier's behavior under systematic uncertainties, we examine the distribution of predicted signal-class probabilities for all four physical processes. Figure 5 shows these distributions both without and with yield weighting. In the left panel, each simulated event counts once (unweighted), directly representing the classifier output for each event. It highlights the intrinsic separation between signal and background classes achieved by the model: signal events ($H \to \tau\tau$) tend to cluster near high probability values, while background processes including $t\bar{t}$, $Z \to \tau\tau$ and diboson dominate the low-probability region.

In the right panel, each simulated event is assigned a per-event weight such that, the total of each process equals its expected yield (cross section × luminosity); thus the curves represent expected event counts rather than raw sample frequencies. The horizontal axis is the classifier's predicted probability for the signal class; larger values indicate more signal-like events. Importantly, this plot accounts for systematic uncertainties by evaluating expected event counts on a three-dimensional grid in ($\alpha_{\text{tes}}, \alpha_{\text{jes}}, \alpha_{\text{met}}$) with size $17 \times 17 \times 41$, i.e. $17 \cdot 17 \cdot 41 = 11849$ fixed nuisance-parameter configurations (uniformly covering $\alpha_{\text{tes}}, \alpha_{\text{jes}} \in [0.96, 1.04]$ and $\alpha_{\text{met}} \in [0, 5]$); see Section 4.2 for construction details. For each bin of the output probability, the solid lines denote the average expected yield across all nuisance points, while the shaded bands capture the full variation envelope — bounded by the maximum and minimum bin counts observed across the parameter grid. The narrowness of the shaded bands indicates that the classifier outputs remain stable under a wide range of nuisance variations. This behavior confirms that the training strategy—based on loss averaging over stochastic perturbations—effectively suppresses sensitivity to systematic shifts without sacrificing the ability to distinguish signal from background.

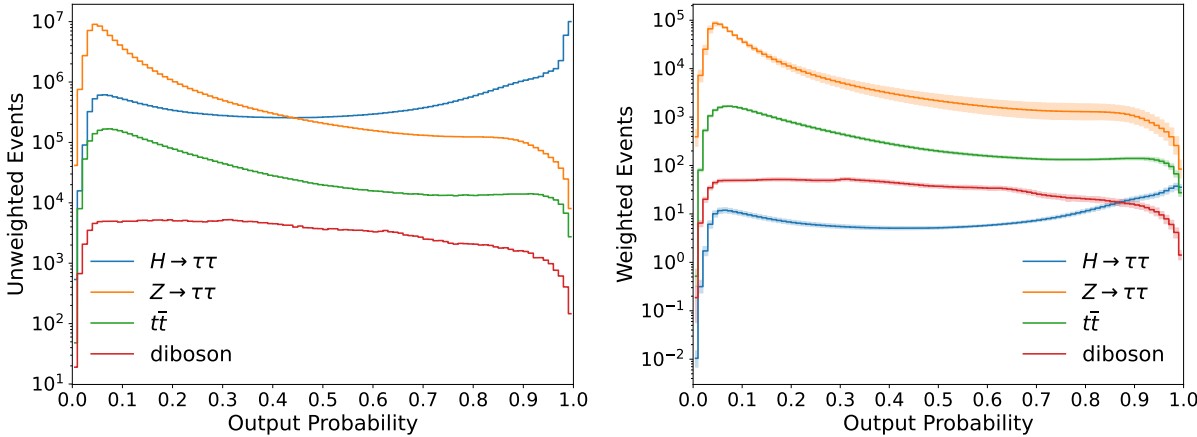

Figure 5: Distributions of the predicted signal-class probabilities for the four processes based on the output of the four-class classifier. The left panel shows the unweighted event distributions, while the right panel includes the expected event yields after applying cross-section-based event weights. In the weighted plot, the solid lines indicate the average count of events per bin evaluated across a grid of 11849 fixed nuisance parameter settings, uniformly spanning $\alpha_{\text{tes}} \in [0.96, 1.04]$, $\alpha_{\text{jes}} \in [0.96, 1.04]$, $\alpha_{\text{met}} \in [0, 5]$. The shaded bands correspond to the envelope formed by the maximum and minimum bin counts over these variations.

## 4 Signal Strength Profiling via Surrogate Likelihood

### 4.1 Region-wise Interpolation and Adaptive Binning

To extract the signal strength $\mu$, we construct a surrogate likelihood from binned event counts of the signal-class probability distributions across multiple analysis regions. For each region, we build a separate interpolation table that maps nuisance-parameter configurations to the process-wise binned yields. As summarized in Table 6, we consider four regions whose selection criteria are taken from Ref. [55]:

The **inclusive region** comprises all events that pass the baseline selection, without imposing any further cuts. It contains approximately 966 expected $H \rightarrow \tau\tau$ events, with a signal-to-background ratio around $10^{-3}$.

The **highMT-VBFJet region** (denoted as CR1 in Table 6) aims to select vector boson fusion (VBF) Higgs production but is depleted of signal due to the high transverse mass requirement. Events are required to have at least two jets with transverse momenta $p_{\text{T}}^{j_1} > 50\,\text{GeV}$ and $p_{\text{T}}^{j_2} > 30\,\text{GeV}$, mimicking the VBF topology. In addition, the transverse mass between the lepton and the missing transverse momentum must satisfy $m_{\text{T}} > 70\,\text{GeV}$. This selection ensures that most signal-like events are removed, while retaining a substantial number of $t\bar{t}$ events, resulting in a signal-to-background ratio of $8.3 \times 10^{-4}$. The region is therefore strongly $t\bar{t}$ dominated and provides an effective constraint on the $\alpha_{t\bar{t}}$ nuisance parameter.

The **highMT-noVBFJet-tt** (denoted as CR2 in Table 6) is designed to enhance $t\bar{t}$-like events that are not already captured by the VBF selection. Events are selected to have high transverse mass ($m_{\text{T}} > 70\,\text{GeV}$) and must veto the VBF topology. Furthermore, it requires that the class output probability of the four-class classifier $\hat{p}_{t\bar{t}}$ exceed 0.4. This requirement ensures that the selected events are strongly $t\bar{t}$-like in nature. In this region, the signal contribution is negligible (only 2.7 events), while the expected background yield from $t\bar{t}$ exceeds 3600, making it highly

effective in constraining the $\alpha_{t\bar{t}}$ nuisance parameter.

The **highMT-noVBFJet-VV** (denoted as CR3 in Table 6) is constructed analogously to CR2, but instead focuses on isolating the diboson background. It also applies the high-$m_T$ cut and vetoes the VBF topology. In this region, the event selection requires that the diboson class output of the four-class classifier $\hat{p}_{VV}$ exceed 0.2. This yields a sample enriched in diboson events, with relatively suppressed contributions from signal and other backgrounds. The expected diboson yield is 597.4, compared to only 1.8 from the signal process and 207.6 from $t\bar{t}$. Although CR3 has the smallest total event count among all regions, its relatively high purity makes it useful for constraining the $\alpha_{VV}$ nuisance parameter.

| Region | Requirements | Type | Poisson yield $\mathcal{L}\sigma$ | | | | $S/B$ |
|---|---|---|---|---|---|---|---|
| | | | $H \to \tau\tau$ | $Z \to \tau\tau$ | $t\bar{t}$ | VV | |
| `inclusive` | – | – | 966.0 | 901 137.5 | 41 283.4 | 3 433.5 | $1.02 \times 10^{-3}$ |
| `highMT-VBFJet` | $p_T^{j_1} > 50$ GeV $p_T^{j_2} > 30$ GeV $m_T > 70$ GeV | CR1 | 14.7 | 721.7 | 16 768.6 | 193.2 | $8.30 \times 10^{-4}$ |
| `highMT-noVBFJet-tt` | $m_T > 70$ GeV veto on VBFJet $\hat{p}_{t\bar{t}} > 0.4$ | CR2 | 2.7 | 202.7 | 3 607.1 | 268.8 | $6.62 \times 10^{-4}$ |
| `highMT-noVBFJet-VV` | $m_T > 70$ GeV veto on VBFJet $\hat{p}_{VV} > 0.2$ | CR3 | 1.8 | 189.5 | 207.6 | 597.4 | $1.8 \times 10^{-3}$ |

Table 6: Summary of the event selections. Control regions are denoted by the "CR" label. Expected Poisson event yields, calculated as $\mathcal{L}\sigma$ with $\mathcal{L}$ representing the integrated luminosity, are provided for each process across different regions. The final column lists the corresponding signal-to-background ratios. Compared to Ref [103], the total yields in the "inclusive" row are slightly reduced due to the stricter threshold requirements on $p_T^{\tau_{had}}$ imposed by the FAIR-HUC benchmark. $\hat{p}_{t\bar{t}}$ and $\hat{p}_{VV}$ refer to the predicted class probabilities for the $t\bar{t}$ and diboson processes, respectively, as assigned by the four-class classifier.

The expected Poisson event yields, computed as $\mathcal{L}\sigma$, span several orders of magnitude across different processes and regions. To accommodate the wide dynamic range and sparse occupancy in some bins, we adopt a region-specific adaptive binning strategy based on the distribution of the signal class output probability of the four-class classifier. The binning process begins with 100 uniformly spaced bins. The neighbouring bins are then iteratively merged in a greedy manner until each remaining bin contains at least ten effective weighted events from the $H \to \tau\tau$ signal process and all three major backgrounds: $Z \to \tau\tau$, $t\bar{t}$, and diboson. This procedure is applied independently for each region. As a result, the inclusive region ends up with 52 bins after merging, while CR1, CR2, and CR3 contain 14, 15, and 13 bins, respectively. This region-wise adaptive binning guarantees that the interpolated event yields are statistically meaningful and stable under systematic variations, while preserving the discriminating power in signal-sensitive regions.

## 4.2 Parametrisation of Systematics and Event Rate Interpolation

To ensure smoothness and avoid boundary effects in the likelihood surface, we reparameterize $\alpha$ in terms of an unconstrained vector of standard normal variables $\nu = (\nu_{tes}, \nu_{jes}, \nu_{met}, \nu_{t\bar{t}}, \nu_{VV}, \nu_{bkg})$, such that each component of $\nu$ follows a Gaussian prior centered at zero. The mapping from $\nu$

to $\alpha$ is defined as:

$$\alpha_{\text{tes}} = 1 + 0.01 \cdot \nu_{\text{tes}},$$
$$\alpha_{\text{jes}} = 1 + 0.01 \cdot \nu_{\text{jes}},$$
$$\alpha_{\text{met}} = e^{\nu_{\text{met}}} - 1,$$
$$\alpha_{t\bar{t}} = e^{\sigma_{t\bar{t}} \nu_{t\bar{t}}},$$
$$\alpha_{VV} = e^{\sigma_{\text{VV}} \nu_{VV}},$$
$$\alpha_{\text{bkg}} = e^{\sigma_{\text{bkg}} \nu_{\text{bkg}}}.$$

where the prior widths are set to $\sigma_{t\bar{t}} = 0.02, \sigma_{\text{VV}} = 0.25, \sigma_{\text{bkg}} = 0.001$, reflecting the expected scale of variation in each process.

To propagate the effect of calibration uncertainties to the event yields, we construct dense three-dimensional interpolation grids in the physical space of $(\alpha_{\text{tes}}, \alpha_{\text{jes}}, \alpha_{\text{met}})$ using simulated predictions at fixed systematic shifts. Each region is associated with a table of shape $17 \times 17 \times 41$, uniformly covering a hypercube extending to $\pm 4\sigma$ along each axis. At every grid point, we store the binned raw event yields for four individual processes: the signal ($S_{\text{raw}}$) and the three main backgrounds ($Z_{\text{raw}}, t\bar{t}_{\text{raw}}, VV_{\text{raw}}$). The total number of grid nodes per region is 11849.

Given a continuous nuisance configuration $\nu$, the corresponding physical scale factors $\alpha$ are computed and used to interpolate event rates via trilinear interpolation:

$$(\alpha_{\text{tes}}, \alpha_{\text{jes}}, \alpha_{\text{met}}) \longrightarrow \{S_{\text{raw}}, Z_{\text{raw}}, t\bar{t}_{\text{raw}}, VV_{\text{raw}}\}, \tag{11}$$

where $S_{\text{raw}}, Z_{\text{raw}}, t\bar{t}_{\text{raw}}, VV_{\text{raw}}$ denote the interpolated templates for signal, $Z \to \tau\tau$, $t\bar{t}$, and diboson processes, respectively. This interpolation is performed independently for each analysis region, using region-specific bin edges defined by the adaptive binning procedure in Section 4.1.

The raw yields are then rescaled using the normalization parameters to construct the total expected counts in each bin. For the Full Region, the expected count in bin $b$ is given by:

$$\lambda_b^{\text{full}}(\mu, \alpha) = \mu \cdot S_{\text{raw}}(\alpha_{\text{tes}}, \alpha_{\text{jes}}, \alpha_{\text{met}}) + \alpha_{\text{bkg}} \cdot Z_{\text{raw}}(\alpha_{\text{tes}}, \alpha_{\text{jes}}, \alpha_{\text{met}})$$
$$+ \alpha_{t\bar{t}} \cdot \alpha_{\text{bkg}} \cdot t\bar{t}_{\text{raw}}(\alpha_{\text{tes}}, \alpha_{\text{jes}}, \alpha_{\text{met}}) + \alpha_{VV} \cdot \alpha_{\text{bkg}} \cdot VV_{\text{raw}}(\alpha_{\text{tes}}, \alpha_{\text{jes}}, \alpha_{\text{met}}),$$

In the three control regions, the signal yield is omitted, and only background components are retained:

$$\lambda_b^{\text{CR}}(\mu, \alpha) = \alpha_{\text{bkg}} \cdot Z_{\text{raw}}(\alpha_{\text{tes}}, \alpha_{\text{jes}}, \alpha_{\text{met}}) + \alpha_{t\bar{t}} \cdot \alpha_{bkg} \cdot t\bar{t}_{\text{raw}}(\alpha_{\text{tes}}, \alpha_{\text{jes}}, \alpha_{\text{met}}) + \alpha_{VV} \cdot \alpha_{bkg} \cdot VV_{\text{raw}}(\alpha_{\text{tes}}, \alpha_{\text{jes}}, \alpha_{\text{met}}),$$

Finally, the bin-wise predictions from all four regions are concatenated into a single event rate vector $\lambda(\mu, \nu)$, which serves as the input to the likelihood function described in Section 4.3. This construction allows the likelihood to respond smoothly to variations in both the signal strength and the nuisance parameters, enabling reliable profiling and uncertainty estimation. Since using a fully parametrized classifier that explicitly depends on $\nu$ would create a severe bottleneck during profiling—where the likelihood requires repeated evaluations across $(\mu, \nu)$—we instead adopt a single classifier that is valid across all nuisance configurations. To make this feasible, the classifier is trained to be robust against nuisance variations, such that any remaining $\nu$-dependence can be handled efficiently through the interpolation tables.

## 4.3  Construction of the Profile Likelihood

To extract the best-fit value of the signal strength parameter $\mu$ and quantify its associated uncertainty, we adopt the profile likelihood formalism. This approach enables the systematic treatment of nuisance parameters $\alpha$, reparameterized in terms of $\nu$, by profiling them out

during inference. The profile likelihood is constructed by minimizing the negative log-likelihood (NLL) with respect to the nuisance parameters at fixed signal strength $\mu$:

$$\text{NLL}_{\text{prof}}(\mu) = \min_{\nu} \text{NLL}(\mu, \nu). \tag{12}$$

The full likelihood function is constructed as a product of independent Poisson terms over all analysis bins, together with Gaussian priors on each component of $\nu$:

$$\mathcal{L}(\mu, \nu) = \prod_{b=1}^{B} \text{Poisson}(n_b(\nu) \mid \lambda_b(\mu, \nu)) \cdot \prod_{i=1}^{6} \mathcal{N}(\nu_i \mid 0, 1), \tag{13}$$

where $n_b(\nu)$ is the observed count in bin $b$, and $\lambda_b(\mu, \nu)$ denotes the expected yield as defined in Section 4.2. In log space, the NLL becomes:

$$\text{NLL}(\mu, \nu) = \sum_{b=1}^{B} [\lambda_b(\mu, \nu) - n_b(\nu) \log \lambda_b(\mu, \nu)] + \frac{1}{2} \|\nu\|^2 + \text{const.} \tag{14}$$

The second term penalizes large deviations of the nuisance parameters from their nominal values via a standard normal prior, enforcing a regularized fit.

## 4.4 Signal Strength Estimation and Confidence Interval Extraction

To determine the most likely value of $\mu$, we first identify the global minimum of the profile NLL:

$$\hat{\mu} = \operatorname*{argmin}_{\mu} \text{NLL}_{\text{prof}}(\mu). \tag{15}$$

Uncertainty estimation is then based on the profile likelihood ratio:

$$\Delta \text{NLL}(\mu) = \text{NLL}_{\text{prof}}(\mu) - \text{NLL}_{\text{prof}}(\hat{\mu}), \tag{16}$$

which serves as the test statistic for constructing confidence intervals. According to Wilks' theorem, $\Delta \text{NLL}$ follows a $\chi^2$ distribution with one degree of freedom in the asymptotic regime. Therefore, the 68.3% confidence interval on $\mu$ corresponds to $\Delta \text{NLL}(\mu) \leq 0.5$.

The minimization over the nuisance parameters $\nu$ at fixed signal strength $\mu$ is carried out using LBFGS optimizer in PyTorch with a strong Wolfe line search. For the first $\mu$ grid point, we obtain an initial seed via dual annealing and then refine it with LBFGS, while for all subsequent points we warm-start from the previous grid point's profiled solution. To mitigate local minima, 5 additional LBFGS restarts from different initializations through random perturbations are performed and the best solution is retained. Box constraints are enforced by projecting the solution back to the allowed ranges after each LBFGS run. The bounds are $[-5, 5]$ for all components except $\nu_{\text{met}} \in [\log(1.001), \log(6)]$, which ensures that $\alpha_{\text{met}} = e^{\nu_{\text{met}}} - 1 \in [0, 5]$.

The core profile scan is performed on a uniform grid of 100 points in $\mu$ over $[0, 3]$. At each grid point we profile $\nu$ as described above, using warm starts to ensure a smooth and efficient profile curve. To provide stable gradients, the full likelihood evaluation is implemented in PyTorch with automatic differentiation. Once the profile curve is obtained, we refine the maximum-likelihood estimate by fitting a simple quadratic function to the seven points closest to the minimum. This gives a more accurate estimate of $\hat{\mu}$ between the grid points. We then *re-profile* the nuisance parameters at this sub-grid $\hat{\mu}$ to obtain the true minimum $\text{NLL}_{\text{min}}$ and the corresponding profiled parameters.

To determine the 68% confidence interval, we solve for the roots of

$$\Delta \text{NLL}(\mu) = \text{NLL}_{\text{prof}}(\mu) - \text{NLL}_{\text{min}} = 0.5, \tag{17}$$

using Brent's method applied to the scan grid with on-the-fly re-profiling of $\nu$ at each function evaluation. In cases where a valid bracket is not found, cubic spline or quadratic fits are used as fallbacks to ensure stability.

# 5 Performance Validation with Pseudo-Experiments

To quantitatively assess the statistical performance of our profile likelihood framework, we conduct three complementary studies: an Asimov study (Section 5.1) and two sets of Monte Carlo studies (Sections 5.2 and 5.3). The Asimov study uses the Asimov dataset to present the profiled likelihood curve for $\mu$ and the corresponding impact ranking of nuisance parameters, providing a baseline validation of the inference pipeline and parameter sensitivities. The two Monte Carlo studies are designed to evaluate the coverage, interval, and parameter estimation related to the extracted signal strength. The first set of studies (Section 5.2) is based on 9000 pseudo-experiments generated at nine fixed signal strengths $\mu^{\text{true}} = 0.3, 0.6, \ldots, 2.7$, with 1000 pseudo-experiments per point. This setup allows for a direct evaluation of the coverage and the interval width associated with the constructed confidence intervals. The second set (Section 5.3) involves 50000 pseudo-experiments with signal strengths sampled uniformly from a continuous range $[0, 3]$. For each pseudo-experiment, we extract the best-fit values of both $\mu$ and the six nuisance parameters. Together, these studies provide a rigorous and comprehensive validation of the likelihood-based inference procedure under both controlled and fully randomized conditions, and can be performed efficiently in our framework, with the full inference for a single pseudo-experiment taking about 5 minutes on a CPU and 2 minutes on a NVIDIA Tesla V100 GPU.

## 5.1 Asimov Study

**Asimov dataset construction.** In our surrogate-likelihood setup, four Asimov datasets are built, one for each analysis region (inclusive, CR1, CR2, CR3), by querying the region-wise interpolation tables at the nominal calibration point

$$\alpha_\star \equiv (\alpha_{\text{tes}}, \alpha_{\text{jes}}, \alpha_{\text{met}}) = (1, 1, 0),$$

For each process $k \in \{H \to \tau\tau, Z \to \tau\tau, t\bar{t}, VV\}$ and region $R$, we obtain the expected per-bin yields in the signal-class probabilities based on the classifier—using the adaptive binning of Section 4.1—directly from the interpolation tables (Section 4.2) evaluated at $\alpha_\star$. Let $S_{b,R}(\alpha_\star)$, $Z_{b,R}(\alpha_\star)$, $t\bar{t}_{b,R}(\alpha_\star)$, and $VV_{b,R}(\alpha_\star)$ denote the resulting histograms for the signal and three backgrounds, respectively. The Asimov bin contents are then defined by summing the process-wise histograms (i.e., without statistical fluctuations) as

$$n^{\text{A}}_{b,\text{inclusive}}(\mu_A) = \mu_A S_{b,\text{inclusive}}(\alpha_\star) + Z_{b,\text{inclusive}}(\alpha_\star) + t\bar{t}_{b,\text{inclusive}}(\alpha_\star) + VV_{b,\text{inclusive}}(\alpha_\star),$$

and, for the three control regions (where the signal template is omitted by construction),

$$n^{\text{A}}_{b,R} = Z_{b,R}(\alpha_\star) + t\bar{t}_{b,R}(\alpha_\star) + VV_{b,R}(\alpha_\star), \qquad R \in \{\text{CR1, CR2, CR3}\}.$$

We take $\mu_A = 1$ and set all normalization-type nuisance parameters to their nominal values when forming the Asimov dataset.

**Profiled likelihood on Asimov.** Using the four region-wise Asimov histograms, we profile the likelihood as described in Section 4.4). For each grid point we minimize the NLL to obtain

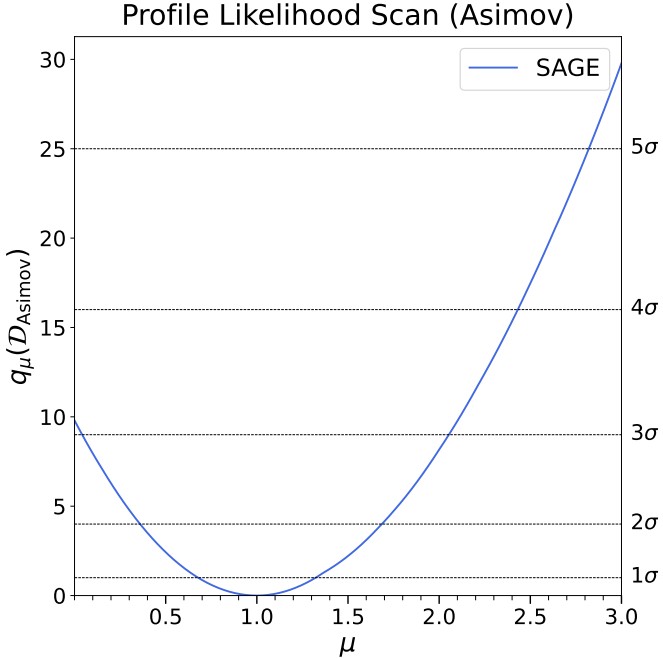

Figure 6: Asimov dataset fit scan of the signal strength parameter $\mu$, with the profile likelihood ratio $q_\mu$ shown as a function of $\mu$.

$\text{NLL}_{\text{prof}}(\mu) = \min_\nu \text{NLL}(\mu, \nu)$, and form the (asymptotic) profile-likelihood test statistic, which differs from Eq. (17) only by a factor of 2:

$$q_\mu(\mathcal{D}_\text{A}) = 2\Big[\text{NLL}_{\text{prof}}(\mu) - \text{NLL}_{\text{prof}}(\hat\mu)\Big], \tag{18}$$

where $\hat\mu$ is the best fit. Under Wilks' theorem for one degree of freedom, $q_\mu$ is approximately $\chi_1^2$-distributed, so the intersections with $q_\mu = 1, 4, 9, 16, 25$ correspond to the asymptotic $1\sigma$–$5\sigma$ levels.

Figure 6 shows the resulting $q_\mu(\mathcal{D}_\text{A})$ curve. As expected, the minimum occurs near $\mu \simeq 1$. We observe no noticeable bias in the location of the minimum: the profiled fit returns $(\hat\mu, \hat\nu) = (1, \mathbf{0})$ with a numerical deviation below $10^{-3}$. Repeating the construction for injected values $\mu_A \in [0.1, 3]$ and re-fitting confirms this behavior.

**Postfit (Asimov).** We start from the Asimov best-fit point $(\hat\mu, \hat\nu)$ obtained at $\boldsymbol{\alpha}_\star = (1, 1, 0)$ with $\mu_A = 1$. For each nuisance $\nu_i$, the *pre-fit* uncertainty is fixed to one standard deviation by the unit-width standard-normal prior. The *post-fit* $1\sigma$ interval is then determined on the Asimov dataset by scanning $\nu_i$ around its best-fit value while profiling all other parameters; the two endpoints are defined by the condition that the profiled likelihood increases by one unit in the usual $\Delta q = 1$ convention. These posterior intervals are consistently narrower than the unit pre-fit widths, reflecting the information gained from the four regions.

**Nuisance-parameter impacts (Asimov).** Impacts on the signal strength are evaluated using the *posterior* $1\sigma$ (not the prior). For each $\nu_i$ we freeze it at $\hat\nu_i \pm \sigma_i^{\text{post}}$, re-profile all remaining nuisance parameters together with $\mu$, and record the corresponding shifts in the best-fit $\mu$. The impact of $\nu_i$ is reported as the larger of the absolute shifts from the two variations. For the one-sided met prior, only the $+1\sigma$ variation is considered both for defining the post-fit interval and for the impact.

Figure 7 shows the pre-fit (unit-width bands) and post-fit ($1\sigma$ error bars from the Asimov profile) for each nuisance parameter, together with the impacts computed as described above.

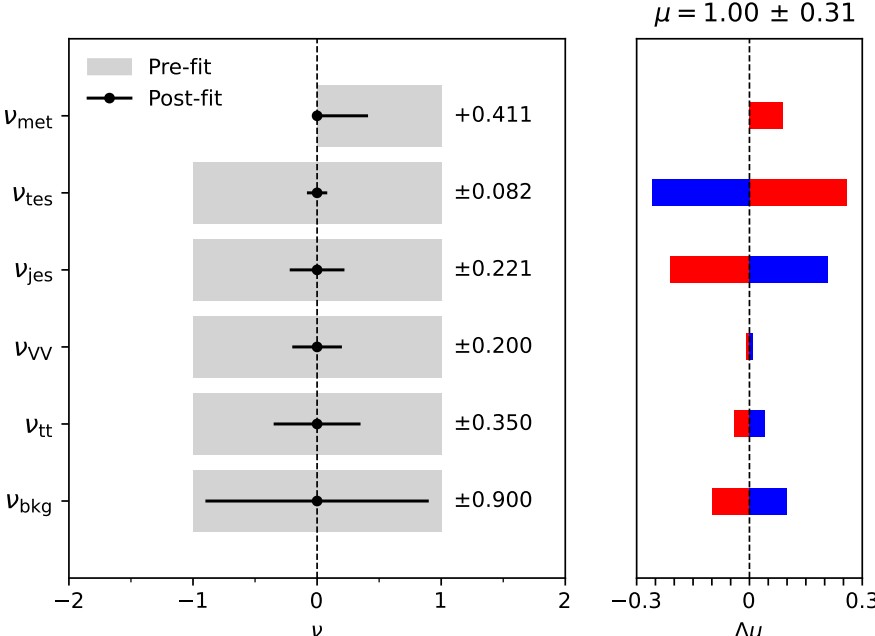

Figure 7: Post-fit and impact plot showing the effect of each nuisance parameter $\nu_i$ on the fitted value of $\mu$, including both pre-fit (shaded bands) and post-fit (error bars) uncertainties. The red (blue) bars on the right quantify the upward (downward) shift in $\mu$ when $\nu_i$ is varied by $+1\sigma$ $(-1\sigma)$.

Calibration nuisance parameters ($\nu_{\text{tes}}$ and $\nu_{\text{jes}}$) are significantly constrained in the post-fit, and normalization nuisance parameters ($\nu_{t\bar{t}}$, $\nu_{\text{VV}}$ and $\nu_{\text{bkg}}$) also exhibit nontrivial reductions; while $\nu_{\text{met}}$ is treated as one-sided in accordance with its prior. Overall, the ranking reflects the constraints provided jointly by the inclusive region and the three control regions in the surrogate-likelihood fit. Ordering the six nuisance parameters by absolute impact on $\mu$ reveals a clear hierarchy: $\nu_{\text{tes}}$ and $\nu_{\text{jes}}$ dominate, followed by $\nu_{t\bar{t}}$ and $\nu_{\text{VV}}$, while $\nu_{\text{bkg}}$ and the one-sided $\nu_{\text{met}}$ have the smallest impacts.

## 5.2  Coverage Evaluation at Fixed Signal Strengths

To evaluate the coverage and interval width of our inference framework, we conduct a large-scale Monte Carlo study based on 9000 pseudo-experiments. Each pseudo-experiment is generated from one of nine fixed signal strengths, $\mu^{\text{true}} = 0.3, 0.6, \ldots, 2.7$, with 1000 pseudo datasets produced for each value. For every pseudo-experiment, we independently sample the six nuisance parameters from their respective priors defined in the FAIR-HUC benchmark.

Using the injected values of $\mu^{\text{true}}$ and $\alpha^{\text{true}}$, a perturbed test dataset is constructed by applying the corresponding systematic shifts to the nominal features, including all derived kinematic and event-level observables.

For each pseudo-experiment, the trained classifier is applied to its events, and the resulting outputs are processed through the profile pipeline described in Section 4. A profile likelihood curve is then constructed by scanning over a fixed $\mu$ grid within the range $[0, 3]$. From this curve, we extract the maximum likelihood estimate $\hat{\mu}$ and construct the 68.3% confidence interval using the profile likelihood ratio criterion $\Delta\text{NLL}(\mu) \leq 0.5$.

The pseudo-experiments in this setup allow us to directly compute the empirical coverage of the constructed confidence intervals by checking how often $\mu^{\text{true}}$ falls within the estimated

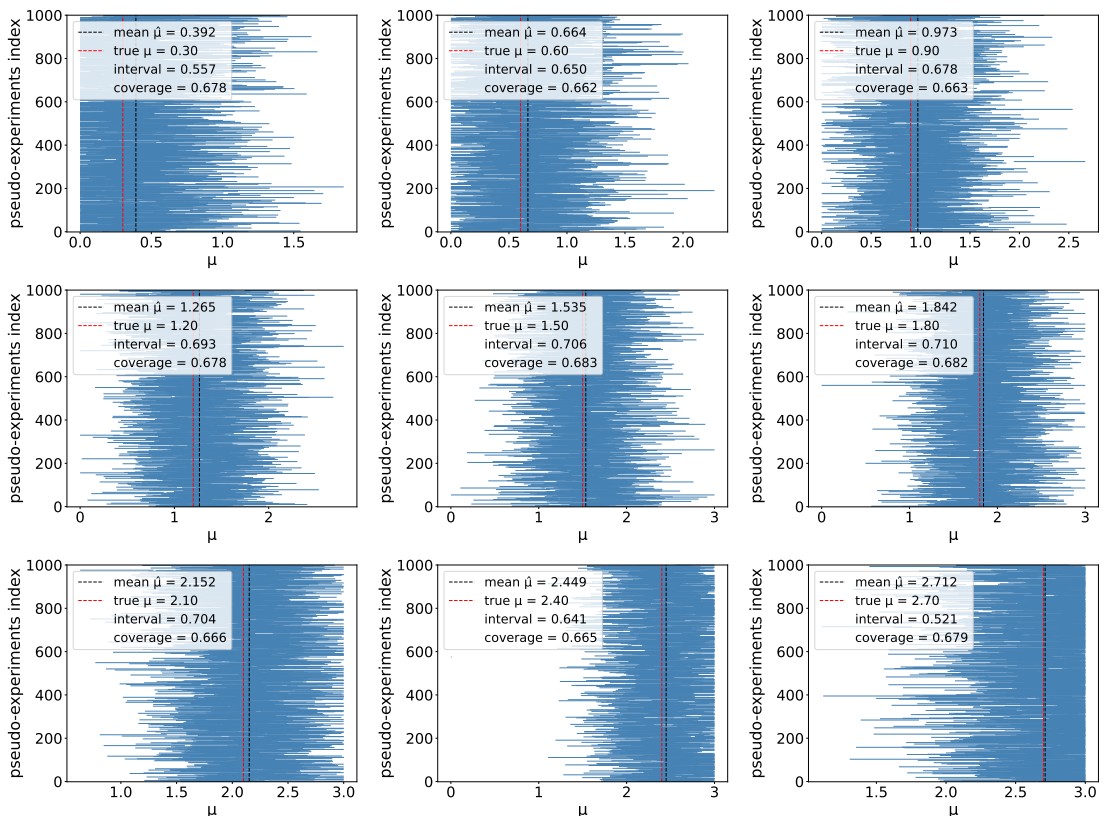

Figure 8: Visualization of 1000 pseudo-experiments for each of the signal strength values $\mu_{\text{true}} = 0.3, 0.6, 0.9, 1.2, 1.5, 1.8, 2.1, 2.4$, and $2.7$ (from top-left to bottom-right). For each pseudo-experiment, the predicted 68.27% confidence interval for $\mu$ is shown as a horizontal blue line. The vertical dotted line indicates the true value $\mu_{\text{true}}$, and the dashed black line marks the average of the maximum likelihood estimates $\hat{\mu}$. The coverage is computed as the fraction of pseudo-experiments where $\mu_{\text{true}}$ lies within the predicted interval.

interval $(\hat{\mu}_{16}, \hat{\mu}_{84})$. We also examine the typical widths of the intervals. These two metrics are quantified as follows:

$$w = \frac{1}{N_{\text{test}}} \sum_{i=1}^{N_{\text{test}}} \left| \mu_{84,i} - \mu_{16,i} \right| \,, \tag{19}$$

$$c = \frac{1}{N_{\text{test}}} \sum_{i=1}^{N_{\text{test}}} \mathbb{I}\left[ \mu_{\text{true},i} \in [\mu_{16,i}, \mu_{84,i}] \right] \,, \tag{20}$$

where $w$ denotes the average interval width and $c$ denotes the empirical coverage. Here, $\mu_{16,i}$ and $\mu_{84,i}$ represent the lower and upper bounds of the 68.3% confidence interval for the $i$-th pseudo-experiment, and $\mathbb{I}[\cdot]$ is the indicator function.

The results shown in Figures 8 demonstrate that the constructed confidence intervals maintain stable performance across the entire range of injected signal strengths. The average interval width increases from $w = 0.557$ at $\mu^{\text{true}} = 0.3$ to a maximum of $w = 0.71$ at $\mu^{\text{true}} = 1.8$, before decreasing to $w = 0.521$ at $\mu^{\text{true}} = 2.7$. This behavior can be attributed to two complementary effects. In the central region, where signal and background contributions are more comparable, statistical uncertainties are larger, leading to wider intervals. Towards the edges of the scan range, the likelihood function is truncated by the fixed bounds at $\mu = 0$ and $\mu = 3$, which

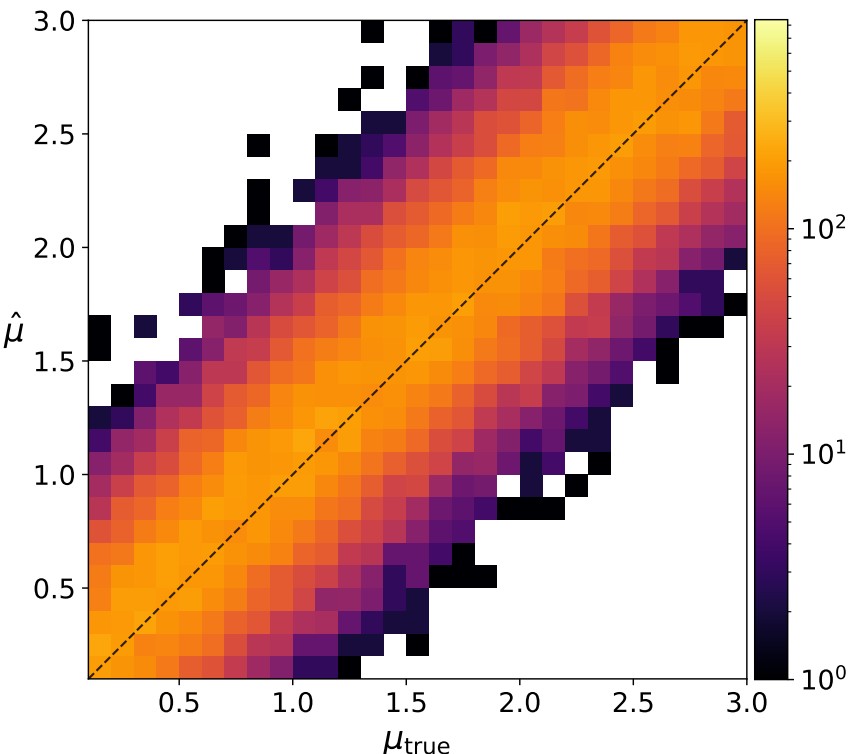

Figure 9: Scatter plot of $\mu_{\text{true}}$ versus the maximum likelihood estimate $\hat{\mu}$ from $5 \cdot 10^4$ pseudo-experiments.

artificially suppresses the width of the resulting intervals. Overall, the interval widths remain relatively narrow across all tested values of $\mu^{\text{true}}$, indicating a high precision in signal strength estimation without sacrificing coverage. The empirical coverage remains close to the nominal 68.3% target throughout, ranging from $c = 0.662$ to $c = 0.683$, with no significant under- or over-coverage observed. The slight fluctuations across different values of $\mu^{\text{true}}$ are consistent with statistical variability at the level of 1000 pseudo-experiments per point. These results confirm that the surrogate likelihood framework provides statistically reliable confidence intervals, with well-controlled coverage properties.

## 5.3  Global Performance under Randomized Signal Strengths

To further validate the robustness and reliability of our inference framework, we conduct a second Monte Carlo study using 50000 pseudo-experiments. In contrast to the fixed-$\mu$ setup in Section 5.2, here the true signal strength $\mu^{\text{true}}$ is drawn uniformly from the continuous range $[0, 3]$, allowing us to probe the global behavior of the profile likelihood across a wide spectrum of scenarios.

Figure 9 shows a scatter plot of $\hat{\mu}$ versus $\mu_{\text{true}}$, with points closely aligned along the diagonal and no significant bias or outliers. This indicates that the surrogate likelihood yields stable and accurate point estimates across the full range of signal strengths, with fluctuations consistent with statistical uncertainty.

Figures 10 display the scatter plots of the maximum likelihood estimates $\hat{\nu}$ versus the injected true values $\nu_{\text{true}}$ for all six nuisance parameters. Overall, the reconstructed values align well with the truth, clustering around the diagonal with very few outliers, which confirms the stability and reliability of the inference procedure across all systematic directions. Among

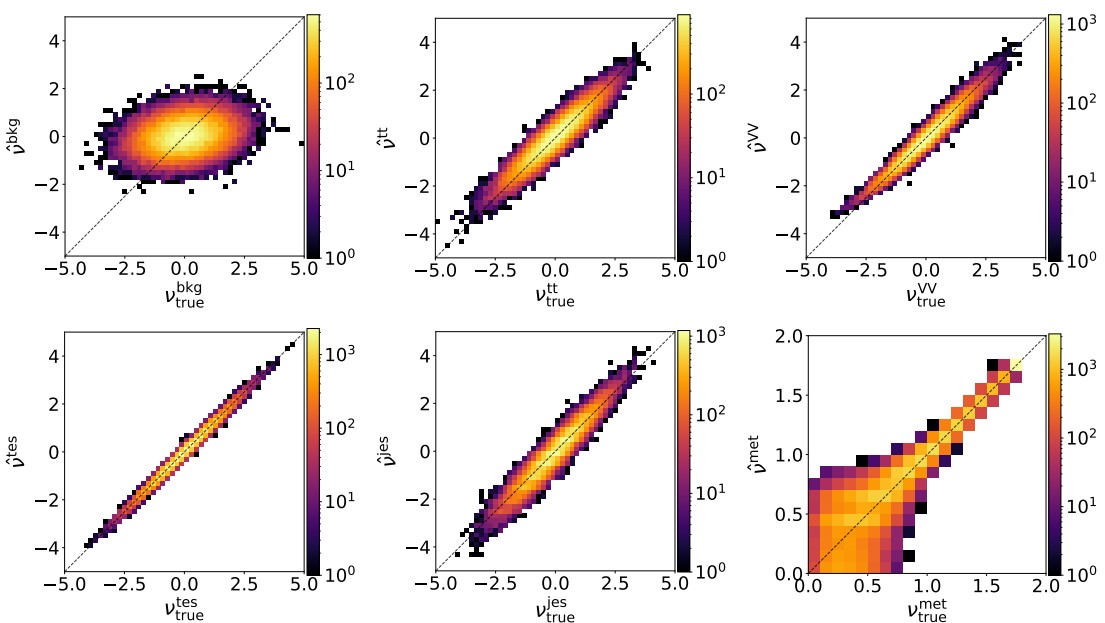

Figure 10: Same as Fig. 9, but for all six nuisance parameters: $\nu_{\text{bkg}}$, $\nu_{t\bar{t}}$, and $\nu_{\text{VV}}$ (top row, left to right), and $\nu_{\text{tes}}$, $\nu_{\text{jes}}$, and $\nu_{\text{met}}$ (bottom row, left to right).

the calibration-type parameters, $\nu_{\text{tes}}$ exhibits the best performance, with tight correlation to the injected values. This superior behavior arises from the presence of a $\tau$-jet transverse momentum threshold at 26 GeV in the original dataset, which makes the total event yield highly sensitive to variations in $\nu_{\text{tes}}$. Consequently, the surrogate likelihood naturally favors accurate adjustment of this parameter during profiling. In comparison, $\nu_{\text{jes}}$ is also well constrained but primarily affects the shape rather than the overall event yield, resulting in slightly less sensitivity and a broader spread than $\nu_{\text{tes}}$. For the normalization-type nuisances, $\nu_{\text{bkg}}$ is poorly constrained and shows little correlation with the true value. This is expected: in the full region, the dominant background process is $Z \rightarrow \tau\tau$, and even large variations in $\nu_{\text{bkg}}$ (up to 5 standard deviations) translate to only about a 0.5% change in total yield. In contrast, $\nu_{t\bar{t}}$ and $\nu_{\text{VV}}$ dominate in their respective control regions and can induce 10% and even 100% variations in event yields, respectively, making them far more identifiable in the profiling process. As a result, $\nu_{\text{VV}}$ is especially well reconstructed, followed by $\nu_{t\bar{t}}$. Finally, $\nu_{\text{met}}$ exhibits slightly degraded performance at the low end of its range, where variations in the parameter induce only negligible shifts in the event yields. In this regime, the profile likelihood becomes nearly flat along the $\nu_{\text{met}}$ direction, making it difficult to determine a precise MLE and leading to the observed bias and increased spread. By contrast, at larger values of $\nu_{\text{met}}$, the same variations result in more pronounced yield changes, which steepen the likelihood curve and allow for a more precise estimation.

Figure 11 presents the average widths of the 68.3 % (red) and 95 % (blue) confidence intervals given by $\mu_{84} - \mu_{16}$ and $\mu_{97.5} - \mu_{2.5}$, obtained from 50000 pseudo-experiments with $\mu_{\text{true}}$ drawn uniformly in $[0,3]$. The shaded bands indicate the corresponding $1\sigma$ spread. The interval width is smallest near the scan boundaries and reaches its maximum around $\mu^{\text{true}} \approx 1.5$. The edge–wise narrowing is a consequence of the fixed scan limits at $\mu = 0$ and $\mu = 3$. The right panel shows the empirical coverage, defined in (19). Both the 68.3% and 95% intervals demonstrate excellent agreement with their nominal targets across the full range, with only small fluctuations consistent with the expected sampling noise from 50000 pseudo-experiments.

On the FAIR Universe Codabench final evaluation (100×100 values of $\mu$, each boot-

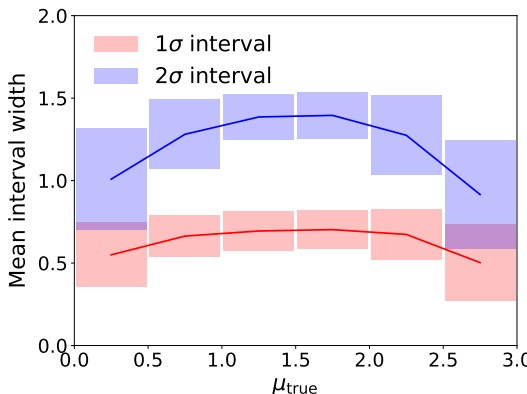 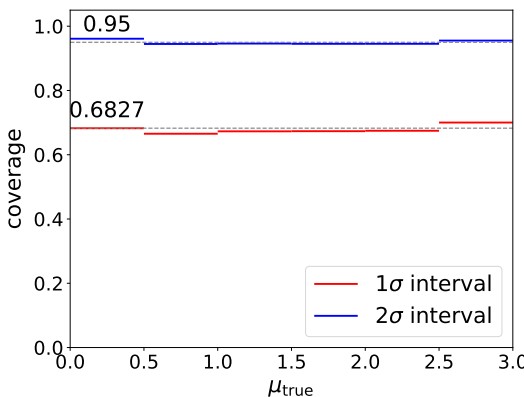

Figure 11: The average width of the confidence interval for the signal strength $\mu$ (left) and the corresponding coverage (right) are shown as functions of the true value $\mu^{\text{true}}$, based on results from 50000 pseudo-experiments. The average interval length is 0.6342 and the coverage is 0.6697.

strapped 1000 times for precision), the two top-performing submissions report average coverages and interval lengths of (coverage, interval) = (0.6683, 0.4599) for Reference [55] and (0.6698, 0.4974) for Reference [100]. Using our protocol (50000 pseudo-experiments with $\mu \sim \mathcal{U}[0,3]$), we obtain an average coverage of 0.6697 and an average interval length of 0.6342. While these protocols are not identical and a like-for-like re-evaluation would be required for a rigorous comparison, the coverage is comparable to the leaderboard methods, with moderately wider intervals. A practical advantage of our approach is its ease of extension to more complex final states: graph-based representations naturally accommodate high object multiplicities and intricate topologies, allowing the model to exploit multi-object correlations that become increasingly salient in such processes.

# 6   Conclusions

We have introduced a dedicated graph neural network framework for Higgs signal strength estimation under systematic uncertainties. The architecture combines a deterministic GNN and an uncertainty-aware GNN, enabling separate treatment of stable and perturbed input features. Systematic variations are explicitly injected during training via dynamic sampling of multiple nuisance parameter configurations in each epoch, encouraging robustness in the classifier outputs. These outputs are propagated through a template morphing pipeline to construct a surrogate likelihood function that supports efficient profile likelihood inference.

Validated on the FAIR-HUC benchmark, our method delivers accurate maximum likelihood estimates of the signal strength $\mu$ and its associated confidence intervals. In large-scale pseudo-experiments, the resulting intervals exhibit excellent coverage and controlled widths across both fixed and randomized signal strengths. The nuisance parameters are consistently and stably reconstructed, with fitted values closely tracking their injected ground truths. In this sense, our framework explicitly operates in the regime of "known unknowns", where the relevant sources of uncertainty are identified but their exact magnitudes must be inferred.

Our publicly available implementation, SAGE [102], provides a flexible and scalable solution for likelihood-based inference, combining deep learning with statistical rigor. The framework is directly applicable to precision measurements at the LHC and readily extensible to other domains where systematic uncertainties are a dominant concern, with particular advantages for processes

involving complex topologies or high object multiplicities where graph-based representations can fully exploit intricate correlations. Looking ahead, we note that the method scales naturally with the number of nuisance parameters: the network performance itself is largely insensitive to the number of nuisance parameters included, although the construction of the interpolation table becomes increasingly challenging. Addressing this will likely require more efficient strategies such as sparse grids or normalizing flows for high-dimensional interpolation.

## A   Re-training with Class-Frequency Weighting

All results in the main text use uniform per-event weights. However, the FAIR-HUC training set is strongly class-imbalanced—the four classes ($H \to \tau\tau$, $Z \to \tau\tau$, $t\bar{t}$, $VV$) differ in event counts by orders of magnitude. As summarized in Table 2, the raw event counts in the FAIR-HUC dataset also differ substantially from the physics-expected event yields based on cross section and luminosity, since the signal class is oversampled. Consequently, the equal-weight scheme used in the main analysis neither corresponds to luminosity-scaled training nor equalizes class contributions.

   In this appendix we evaluate the effect of introducing class weights: we re-train the classifier with the SCIKIT-LEARN "balanced" scheme. For class $k$ with $n_k$ training events, total $N = \sum_k n_k$, and $K = 4$ classes, the class weight is $w_k = \frac{N}{K n_k}$. We then compare the re-trained model to the uniform-weight baseline. The training/validation splits and model hyperparameters are kept unchanged, and the full surrogate-likelihood pipeline (interpolation, profiling, and interval extraction) is repeated. This appendix summarizes the resulting changes and cross-checks: (i) unweighted and weighted signal-class probability distributions under nuisance variations; (ii) confusion matrices comparing the original (equal-weight) and re-trained (class-weighted) classifiers; and (iii) the final profile-likelihood intervals and coverage based on 50000 pseudo-experiments. These results are intended as a comparative study of class weighting and do not supersede the baseline numbers reported in the main text.

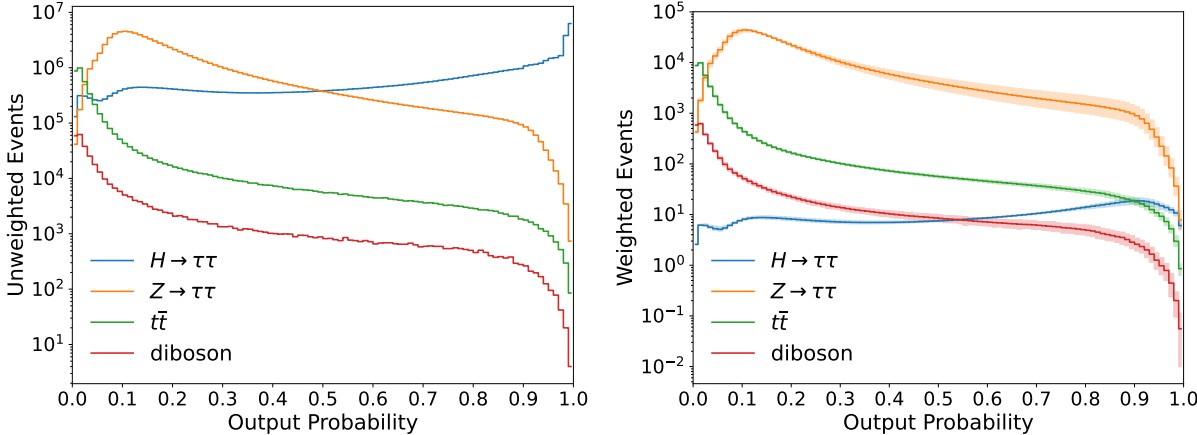

Figure 12: Distributions of the predicted signal-class scores from the re-trained (class-weighted) four-class classifier. Left: unweighted per-event histograms. Right: expected yields after applying per-class weights; curves are averaged over the same grid of 11849 fixed nuisance parameter settings as in the main text, and bands show the min–max envelope. Relative to Fig. 5, the signal shape is nearly unchanged, while $Z \to \tau\tau$ flattens and $t\bar{t}/VV$ become steeper due to class-frequency weighting.

In Fig. 12, we present the distributions of the predicted signal-class probabilities for the four

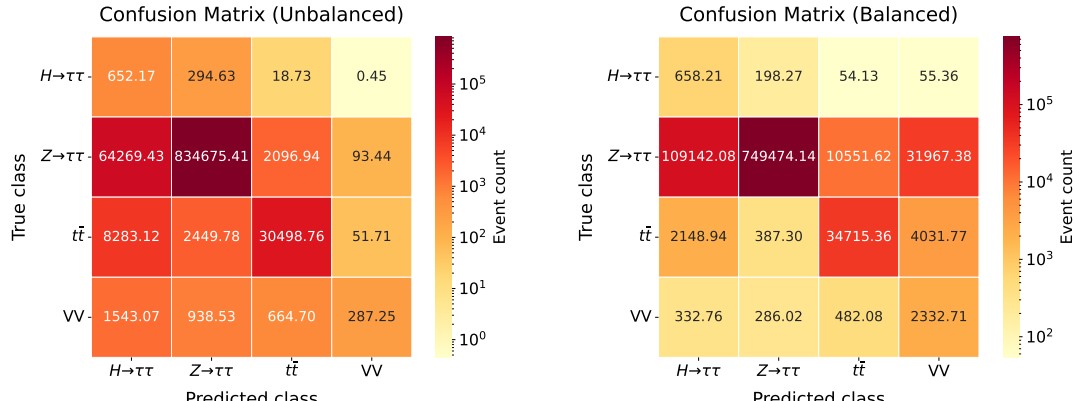

Figure 13: Confusion matrices for two four-class classifiers, shown on a logarithmic color scale. Rows indicate the true class and columns the predicted class; entries are weighted event yields. **Left:** classifier trained with uniform per-event weighting across all classes. **Right:** classifier trained with class-frequency weighting.

processes based on the output of the retrained four-class classifier. Compared to the baseline in Fig. 5, the re-trained classifier (with class-balanced loss) shows only marginal changes for the signal class, whereas the background shapes respond more visibly once process weights are applied (right panel of Fig. 12). In the weighted plot, the $Z \to \tau\tau$ distribution becomes noticeably flatter, while the $t\bar{t}$ and diboson distributions become steeper and concentrate more strongly toward the background-like score region. This pattern follows directly from the class-frequency weighting used during training and in the yield construction: as the dominant background, $Z \to \tau\tau$ receives the smallest weight, whereas the rarer $t\bar{t}$ and diboson processes are up-weighted, boosting their effective contributions and sharpening their weighted profiles. The left panel shows the corresponding unweighted score distributions from the re-trained classifier for reference. As in the main text, the weighted panel accounts for systematics by averaging expected bin counts over 11849 fixed nuisance configurations; the shaded bands indicate the full envelope across this grid and remain narrow, confirming that the score distributions are stable under nuisance variations even after class balancing.

In Fig. 13, we present the confusion matrices for the two four-class classifiers. The vertical axis is the *true class* and the horizontal axis is the *predicted class*; the colour scale is logarithmic. Each cell shows the *weighted* event count (physical event yield) accumulated in that category. The left panel corresponds to the classifier trained with uniform per-event weights, and the right panel to the classifier trained with class-frequency weighting. Comparing the two, the patterns shift as expected from the class weights ($w_{VV} > w_{tt} > w_{H \to \tau\tau} > w_{Z \to \tau\tau}$):

- **$H \to \tau\tau$ row.** The correct classification rate remains similar, while misidentified signal events migrate more often to $t\bar{t}$ and VV, and less often to $Z \to \tau\tau$.

- **$Z \to \tau\tau$ row.** Because $Z \to \tau\tau$ receives the smallest weight, the weighted counts of events misidentified as $H \to \tau\tau/t\bar{t}$/VV increase markedly in the balanced model; the diagonal entry correspondingly decreases in weighted yield.

- **$t\bar{t}$ row.** The weighted number of correctly classified $t\bar{t}$ events increases noticeably; misidentifications into VV also increase, reflecting the up-weighting of both processes.

- **VV row.** With the largest class weight, the correctly classified VV→VV yield grows substantially in the balanced model, while misidentifications into other classes are relatively suppressed in weighted yield.

Overall, class-frequency weighting amplifies the effective contributions of the rarer $t\bar{t}$ and

VV processes and suppresses the dominant $Z \to \tau\tau$, which explains the row-wise changes observed between the two confusion matrices.

Table 7 summarizes the selections and expected yields obtained with the re-trained, class-balanced classifier. The requirements for the `inclusive`, `highMT-VBFJet` (CR1), and `highMT-noVBFJet-tt` (CR2) regions are *identical* to those in Table 6. However, because the classifier decision boundaries shift under class-frequency weighting, the flow of events into the `highMT-noVBFJet-VV` control region (CR3) changes appreciably. In particular, the large class weight assigned to VV increases the weighted misidentification rates into the VV class from the other three processes. To preserve a VV-dominated sample and obtain a tight constraint on the normalization parameter $\nu_{VV}$, we therefore tighten the CR3 classifier requirement to $\hat{p}_{VV} > 0.9$ (last row), while keeping all kinematic cuts unchanged. The table reflects the resulting *weighted* Poisson yields per region and process and the corresponding $S/B$; the inclusive and CR1 yields remain consistent with Table 6, whereas CR2 and CR3 shows the expected shifts due to the updated classifier decision boundaries.

| Region | Requirements | Type | Poisson yield $\mathcal{L}\sigma$ | | | | $S/B$ |
|---|---|---|---|---|---|---|---|
| | | | $H \to \tau\tau$ | $Z \to \tau\tau$ | $t\bar{t}$ | VV | |
| `inclusive` | – | – | 966.0 | 901 137.5 | 41 283.4 | 3 433.5 | $1.02 \times 10^{-3}$ |
| `highMT-VBFJet` | $p_T^{j1} > 50$ GeV $p_T^{j2} > 30$ GeV $m_T > 70$ GeV | CR1 | 14.7 | 721.7 | 16 768.6 | 193.2 | $8.30 \times 10^{-4}$ |
| `highMT-noVBFJet-tt` | $m_T > 70$ GeV veto on VBFJet $\hat{p}_{t\bar{t}} > 0.4$ | CR2 | 3.0 | 304.0 | 3381.5 | 141.9 | $7.96 \times 10^{-4}$ |
| `highMT-noVBFJet-VV` | $m_T > 70$ GeV veto on VBFJet $\hat{p}_{VV} > 0.9$ | CR3 | 3.2 | 464.1 | 198.5 | 725.6 | $2.3 \times 10^{-3}$ |

Table 7: Event selections and expected yields with the re-trained (class-frequency–weighted) four-class classifier. The `inclusive`, `highMT-VBFJet` (CR1), and `highMT-noVBFJet-tt` (CR2) requirements are unchanged from Table 6; CR3 keeps the same kinematic cuts but uses a tighter classifier threshold, $\hat{p}_{VV} > 0.9$, to preserve VV dominance under class weighting. Entries are expected Poisson yields ($\mathcal{L}\sigma$) per process; the last column reports $S/B$. Differences relative to Table 6 reflect shifts in decision boundaries.

In Fig 14, we show the average width of the confidence interval for the signal strength $\mu$ and the corresponding coverage based on retrained classifier from 50000 pseudo-experiments. Relative to the baseline results in Fig. 11, the *coverage* obtained with the class-balanced classifier remains essentially unchanged across the full range of $\mu^{\text{true}}$, with fluctuations compatible with the Monte Carlo statistics. In contrast, the *intervals* are systematically wider. The broadening is driven by how class-frequency weighting reshapes the effective composition of the signal region: after re-training, misidentifications of $t\bar{t}$ and VV into the signal class are reduced, while misidentifications of the dominant $Z \to \tau\tau$ background into the signal class increase. The net effect—visible both in the weighted score distributions of Fig. 12 and in the region yields of Table 7—is a higher background contamination in the signal-enriched region, which reduces the sensitivity to variations in $\mu$ and therefore leads to wider confidence intervals.

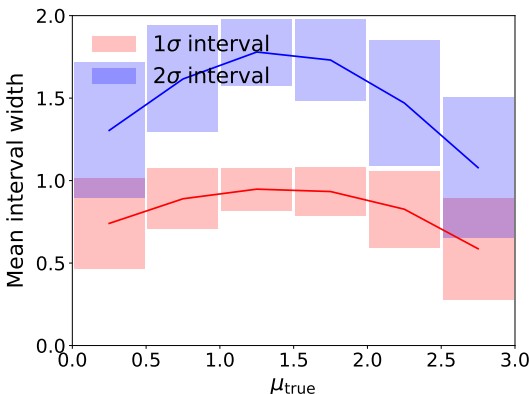 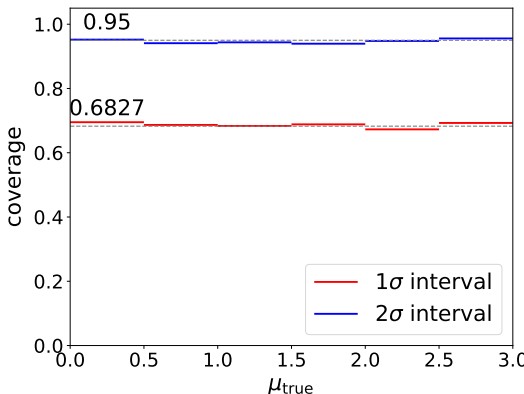

Figure 14: Results obtained with the re-trained (class-frequency–weighted) classifier; the full surrogate-likelihood pipeline matches the main-text setup. Left: average widths of the 68.3% (red) and 95% (blue) intervals as functions of $\mu^{\text{true}}$. Right: empirical coverage of the corresponding intervals. All curves are derived from 50000 pseudo-experiments.

# Acknowledgments

We would like to thank the organizers of the FAIR-HUC challenge for the organization and technical support. We also would like to thank Robert Schöfbeck for fruitful discussions. The computational results were obtained using the CLIP computing cluster of the Austrian Academy of Sciences at https://www.clip.science/.

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
