# Peer review of "Higgs Signal Strength Estimation with Machine Learning under Systematic Uncertainties"

_SciPost Physics_

## Round 1 · Referee Report · Anonymous (Referee 1) · 2025-10-21

Report

Warnings issued while processing user-supplied markup:

  • Inconsistency: Markdown and reStructuredText syntaxes are mixed. Markdown will be used.
    Add "#coerce:reST" or "#coerce:plain" as the first line of your text to force reStructuredText or no markup.
    You may also contact the helpdesk if the formatting is incorrect and you are unable to edit your text.

Dear authors,

Your manuscript presents a robust and technically solid framework for incorporating systematic uncertainties directly into the training and inference of a machine-learning (ML) classifier, specifically using a dual-branch Graph Neural Network (GNN) and a Poisson surrogate likelihood evaluated on a grid of nuisance parameters. The application to the FAIR-HUC H->tautau benchmark and the validation through extensive pseudo-experiments are interesting.

The manuscript is well-structured and tackles a highly relevant and timely challenge in collider physics: integrating ML methods into inference workflows. I would recommend publication after the following comments and suggestions are addressed.

Physics Aspects

  • Scalability of Systematic Parameters: You currently focus on TES, JES, and MET scale variations. While these are important, they are not exhaustive (e.g., luminosity normalization, theoretical scale variations, etc..). It would be helpful to discuss briefly how the framework generalizes to additional sources. The 17x17x41 interpolation grid is already large; please comment on the computational scaling to a higher-dimensional nuisance space and what steps you might suggest to manage the computational cost.
  • Control Region Definition: The likelihood fit uses regions defined primarily using the classifier's output. I would find important to to study the stability of your results if the control regions were defined purely via kinematic cuts (i.e., independent of the GNN score).
  • Systematic-Feature Partitioning: The split between "deterministic" and "uncertainty-aware" features is a key concept, but in realistic analyses, few (none?) features are truly unaffected by systematics. You need to demonstrate the robustness of the method to this partition. A simple test would be to move one or two of your key "deterministic" features (e.g., ΔR, angular separations) into the uncertainty branch and quantify the impact on performance and coverage.
  • Comparison to Classical Treatment: To clearly quantify the benefit of your proposed approach, a key comparison is needed. I would suggest to compare the performance (S/B discrimination and total uncertainty on mu) using only the deterministic branch, trained in the same way, but followed by a traditional template-morphing treatment of the systematics in the final likelihood fit. This will show the true gain of your systematic-aware training versus the traditional approach.
  • The coverage reported is near 68%, but with slightly wider intervals than some leaderboard entries. Please discuss explicitly whether this increased width is an expected consequence of the replica-averaged training (a more conservative estimate) or if it's introduced by the interpolation procedure.
  • Figure 7 (left): Why is the nuisance parameter range for MET shown as one-sided in this figure?

Machine Learning Aspects

  • It is important to understand where the gain in robustness truly comes from. I therefore suggest the following tests: (a) A single-branch GNN trained with the same replica-averaged scheme. (b) Your full dual-branch GNN, but with the gating/attention mechanism removed from the uncertainty branch. Is the gain coming from the architectural split, the replica averaging, or the attention mechanism?
  • Following up on the physics comment, you should test the most extreme case: What happens if all input features are treated as uncertain?
  • The choices for the number of nuisance parameter replicas per batch (100) and the grid density (17x17x41) seem like tuning choices. Could you comment how the performance depends on the choice?
  • The attention and gating mechanisms are interesting and I believe a visualization of the attention weights for a few characteristic nuisance configurations would provide insight into where the network puts more focus.

Recommendation

Ask for major revision

---

## Editorial Decision

awaiting_resubmission